



# Pesticide fate at catchment scale: conceptual modelling of stream CSIA data

Stefanie R. Lutz[1], Ype van der Velde[2], Omniea F. Elsayed[3], Gwenaël Imfeld[3], Marie Lefrancq[3], Sylvain Payraudeau[3], Boris M. van Breukelen[4]

5  [1]UFZ Helmholtz Centre for Environmental Research, Department Catchment Hydrology, Theodor-Lieser-Str. 4, 06120 Halle (Saale), Germany
[2]Department of Earth Sciences, Faculty of Earth and Life Sciences, VU University Amsterdam, De Boelelaan 1085, 1081 HV Amsterdam, The Netherlands
[3]Laboratoire d'hydrologie et de Géochimie de Strasbourg (LHyGeS), Université de Strasbourg/ENGEES, 1 rue Blessig,
10  67084 Strasbourg cedex, France
[4]Delft University of Technology, Faculty of Civil Engineering and Geosciences, Department of Water Management, Stevinweg 1, Delft, The Netherlands

Correspondence to: Stefanie R. Lutz (stefanie.lutz@ufz.de)



**Abstract.** Compound-specific stable isotope analysis (CSIA) has proven beneficial in the characterization of contaminant degradation in groundwater, but it has never been used to assess pesticide transformation at catchment scale. This study presents concentration and carbon CSIA data of the herbicides S-metolachlor and acetochlor from three locations (plot, drain, and catchment outlets) in a 47-ha agricultural catchment (Bas-Rhin France). Herbicide concentrations at the catchment outlet were highest (62 µg L$^{-1}$) in response to an intense rainfall event following herbicide application. Increasing δ$^{13}$C-values of S-metolachlor and acetochlor by more than 2 ‰ during the study period indicated herbicide degradation. To assist the interpretation of these data, discharge, concentrations and δ$^{13}$C-values of S-metolachlor were modelled with a conceptual mathematical model using the transport formulation by travel time distributions. Testing of different model setups supported the assumption that degradation half-lives (DT50) increase with increasing soil depth, which can be straightforwardly implemented in conceptual models using travel time distributions. Moreover, model calibration yielded an estimate of a field-integrated isotopic enrichment factor as opposed to laboratory-based assessments of enrichment factors in closed systems. Thirdly, the Rayleigh equation commonly applied in groundwater studies was tested by our model for its potential to quantify degradation at catchment scale. It provided conservative estimates on the extent of degradation as occurred in stream samples. However, largely exceeding the simulated degradation within the entire catchment, these estimates were not representative of overall degradation at catchment scale. The conceptual modelling approach thus enabled us to upscale sample-based CSIA information on degradation to the catchment scale. Overall, this study demonstrates the benefit of combining monitoring and conceptual modelling of concentrations and CSIA data, and advocates the use of travel time distributions for assessing pesticide fate and transport at catchment scale.

## 1 Introduction

Diffuse pollution of groundwater and rivers is a recurrent issue in agricultural catchments due to the extensive application of pesticides to arable land. Pesticide degradation at catchment scale removes pesticides from the environment, which, provided that pesticide transformation products are non-toxic, reduces their potential impact on the ecosystem. However, pesticide concentrations vary not only due to degradation, but also depending on, e.g., the amount and timing of pesticide application (Battaglin and Goolsby, 1999) or the extent of dilution by pristine water (Schreglmann et al., 2013). Hence, concentration data alone cannot conclusively allow distinction between destructive (i.e., degradation) and non-destructive processes (e.g., transport and sorption). Similarly, laboratory studies allow studying specific mechanisms of pesticide degradation but hardly reflect conditions of pesticide degradation under field conditions (Fenner et al., 2013). These limitations may be overcome by compound-specific isotope analysis (CSIA), which measures the isotopic composition of the contaminant (i.e., the abundance of heavy isotopes relative to light isotopes of an element contained in the compound). The isotopic composition may change under the influence of contaminant transformation (i.e., isotope fractionation; Elsner, 2010; Schmidt and Jochmann, 2012). In contrast, non-destructive processes such as dispersion or sorption may lead to significant isotope fractionation effects only under specific conditions (Eckert et al., 2013; van Breukelen and Prommer, 2008; van Breukelen





and Rolle, 2012). Therefore, CSIA allows for the detection and even quantification of contaminant degradation in polluted environmental systems.

CSIA has been previously applied to study in situ degradation of organic groundwater contaminants (Blum et al., 2009; Elsner et al., 2012; Hunkeler et al., 2005; Schmidt and Jochmann, 2012; Wiegert et al., 2012; Zwank et al., 2005). In the context of diffuse agricultural pollution, CSIA has mainly been used to distinguish natural from anthropogenic nitrate sources, and discern denitrification (Divers et al., 2014; Johannsen et al., 2008; Kellman and Hillaire-Marcel, 2003; Voss et al., 2006; Wexler et al., 2014). Although CSIA may confirm the occurrence of pesticide degradation (Fenner et al., 2013), CSIA data of pesticides remain restricted to the analysis of isotope fractionation under laboratory conditions (Hartenbach et al., 2008; Meyer and Elsner, 2013; Meyer et al., 2009; Penning et al., 2010; Reinnicke et al., 2011; Wu et al., 2014), and grab samples of groundwater and streamwater (Milosevic et al., 2013; Schreglmann et al., 2013). Degradation of chloroacetanilide herbicides and associated isotope fractionation have been recently studied in lab-scale wetlands (Elsayed et al., 2014), but CSIA of herbicides has not yet been applied at catchment scale to evaluate in situ degradation of pesticides. This study presents the first field CSIA-data of pesticides in surface runoff and streamwater from an agricultural catchment. It discusses concentration and carbon CSIA data of two chloroacetanilide herbicides (S-metolachlor and acetochlor) in a 47-ha agricultural catchment (Bas-Rhin, France) at three different locations (i.e., at the plot, drain, and catchment outlet).

In groundwater studies, CSIA-based degradation assessments have been performed by two approaches: the Rayleigh equation and reactive transport modelling. The Rayleigh equation links the measured isotope fractionation effect, via the isotope fractionation factor, to the extent of degradation (Elsner and Imfeld, 2016; Mariotti et al., 1981; Rayleigh, 1896). CSIA data and associated isotope fractionation effects have more recently been simulated using reactive transport models to characterize groundwater pollution (Atteia et al., 2008; D'Affonseca et al., 2011; Pooley et al., 2009; Prommer et al., 2009; van Breukelen et al., 2005; Wanner et al., 2012) and pesticide pollution at hillslope scale via a virtual experimental approach (Lutz et al., 2013). These models demonstrated that the Rayleigh equation systematically underestimates the extent of degradation as occurred in the analysed samples (Abe and Hunkeler, 2006; van Breukelen and Prommer, 2008; van Breukelen and Rolle, 2012).

There is a pressing need to develop reactive transport models simulating CSIA data at catchment scale in order to advance the interpretation of field isotope data of agrochemicals (Elsner and Imfeld, 2016). Therefore, the purpose of this study was to present a conceptual hydrological two-compartment model (i.e., parsimonious mathematical model) that describes pesticide transport, degradation and associated isotope fractionation in the study catchment to identify the dominant processes affecting herbicide fate and transport. The model applies the transport formulation by travel time distributions and thus aims at reconciling hydrological models with water quality models at catchment scale (Hrachowitz et al., 2016). Conceptual flow and transport modelling with travel time distributions has been recently applied at the agricultural catchment scale to simulate atrazine and chloride transport (Van der Velde et al., 2010; Benettin et al., 2013; Botter et al., 2011; Harman, 2015). The novelty of this study lies in the inclusion of CSIA data in such modelling approaches.





Finally, by comparing the model and the Rayleigh equation, this study investigates how to upscale sample-based CSIA information for degradation assessment at catchment scale. Overall, the main objectives of this study are to i) analyse herbicide CSIA data from an agricultural catchment at different scales, ii) develop a quantitative model using travel time distributions for the interpretation of pesticide concentrations and CSIA data at catchment scale, and iii) evaluate the added value of this modelling approach for the assessment of pesticide transport and degradation at catchment scale.

## 2 Methods

### 2.1 Field site description

The study was conducted in a 47-ha headwater catchment, located 30 km north of Strasbourg (Bas-Rhin, France). The catchment characteristics have been previously described in Lefrancq et al. (2017). Briefly, the mean annual temperature between 2005 and 2011 was 11.7 °C, and mean annual precipitation and potential evapotranspiration were 704 mm (±151 mm) and 820 mm (±28 mm), respectively (data from Meteo France station in Waltenheim sur Zorn at 7 km distance from the catchment). Arable land (with corn and sugar beet as main crops) comprises 88 % of the catchment area, with the remainder being roads and patches of grass. Elevation ranges between 190 m and 230 m, and the mean catchment slope is 6.7 % (±4.8 %). The main soil types are calcareous brown earth and calcic soils on hillsides, and colluvial calcic soils in the central thalweg. Soil characteristics and grain size distribution in surface soil were measured by 30 samples in the top 20 cm (clay 30.8±3.9 %, silt 61.0±4.5 %, sand 8.5±4.2 %, CaCO3 1.1±1.6 %, organic matter 2.16±0.3 %, pH KCl 6.7±0.8, phosphorus 0.11±0.04 g kg$^{-1}$, and CEC 15.5±1.3 cmol$^+$ kg$^{-1}$). Additionally, six 2-m profiles were taken, which showed a rapid decrease of organic matter with depth, from about 2.5 % on average in the top 30 cm to about 0.6 % on average at 100 to 150 cm depth. Soil characteristics were assumed homogeneous in the study area for the following analyses.

The catchment is drained by an artificial drainage network of unknown size; at least one drainpipe was active during the study and continuously discharged into the ditch close to the catchment outlet (Fig. 1).

### 2.2 Study compounds

This study considers the two chloroacetanilide herbicides metolachlor (2-chloro-N-(2-ethyl-6-methylphenyl)-N-(2-methoxy-1-methylethyl)acetamide) and acetochlor (2-chloro-N-(ethoxymethyl)-N-(2-ethyl-6-methylphenyl)acetamide; see section S1 and Table S1 in the supplementary material, SM). Metolachlor and acetochlor are commonly applied pesticides (Grube et al., 2011) mainly used for pre-emergence weed control. Both herbicides have been applied in the study catchment since the 1990s. In 2012, S-metolachlor (i.e., the herbicidally active S-enantiomer of metolachlor) was applied on bare soil as spray containing the commercial formulations Mercantor Gold, Dual Gold, or Camix (Syngenta). Acetochlor was applied as the commercial formulation Harness (Dow Agrosciences). According to a survey conducted among local farmers, 10.4 kg of acetochlor and 10.9 kg of S-metolachlor were applied in the catchment mainly in the first two weeks of May 2012. At the





experimental plot, only S-metolachlor was applied (on April 12 and May 1). We will present monitoring results for both S-metolachlor and acetochlor, but focus on S-metolachlor for the modelling.

## 2.3 Monitoring setup

Discharge and concentrations of S-metolachlor and acetochlor were measured between March and August 2012 at three
different scales (plot experiment, drain outlet and catchment outlet). At the catchment outlet, discharge was continuously measured using a Doppler flowmeter (2150 Isco), and flow-proportional samples were taken every 20 m$^3$ with a refrigerated automatic sampler (Isco Avalanche). For the plot experiment, 77.2 m$^2$ were isolated on a sugar beet field with a 60 cm high shield to a depth of 30 cm below the soil surface. Surface runoff exclusively was collected in a polyethylene gutter, and discharge was measured using a Venturi channel combined with a surface water level sensor (ISMA). Flow-proportional
water samples at the plot were taken every 7 L with a refrigerated automatic sampler (Isco Avalanche). Weekly grab water samples were collected from the drain outlet. Lefrancq et al. (2017) provides a more detailed description of the monitoring setup.

## 2.4 Concentration and CSIA analysis

Quantification and CSIA of S-metolachlor and acetochlor are described in detail elsewhere (Elsayed et al., 2014). Briefly,
1 L water samples were filtered, extracted by solid-phase extraction, concentrated under nitrogen flux to one droplet, and resuspended in 500 μL dichloromethane (DCM). Dissolved herbicides were quantified with a GC-MS/MS system with a mean uncertainty of 8 % and quantification limits of 0.05 and 0.02 μg L$^{-1}$ for acetochlor and S-metolachlor, respectively. Herbicide concentrations were determined for 10 samples at the plot, 16 samples at the drain outlet, and 34 samples at the catchment outlet.

Carbon isotope ratios were measured in triplicates with a GC-C-IRMS system. A series of standards was dissolved in DCM to concentrations of 88.1, 35.2, 26.4, 17.6 and 3.5 μM for metolachlor resulting in a corresponding range of signal amplitudes between 120 and 7000 mV. Despite a lower reproducibility for smaller amplitude signals, the obtained values were always within 0.5 ‰ of the averaged $\delta^{13}$C-value for the two compounds. No effect of the SPE-concentration procedure on the analytical precision could be observed.

Carbon isotope ratios (($^{13}$C/$^{12}$C)$_{Sample}$) are reported in per mil (‰) relative to the VPDB (Vienna Pee Dee Belemnite) standard ratio (($^{13}$C/$^{12}$C)VPDB = 0.0112372):

$$\delta^{13}C = \left( \frac{\left(\frac{^{13}C}{^{12}C}\right)_{Sample}}{\left(\frac{^{13}C}{^{12}C}\right)_{VPDB}} - 1 \right) \qquad (1)$$



Carbon isotope ratios of S-metolachlor were obtained for five samples at the plot and six samples at the catchment outlet (between one and nine weeks after the main application day). Additionally, the $\delta^{13}$C-value of one S-metolachlor sample from the application tank used in the plot experiment was determined. Carbon isotope ratios of acetochlor were obtained for three samples at the plot and five samples at the catchment outlet (between two and six weeks after the main application day). No $\delta^{13}$C-values of either herbicide were obtained for the drain outlet due to concentrations of below 0.5 μg L$^{-1}$.

## 2.5 Hydrological model and travel time distributions

The conceptual hydrological model comprises two storage reservoirs: a source zone reservoir representing the upper soil layer onto which the pesticide is applied, and a lower transport zone comprising both unsaturated soil and groundwater (Fig. 2; cf. Benettin et al., 2013; Bertuzzo et al., 2013). The source zone reservoir ($S_{sz}$) is fed by precipitation ($P$) and the transport zone ($S_{tz}$) is fed by recharge from the source zone reservoir ($R_{tz}$). Water in the source zone storage can leave the storage as evapotranspiration ($ET_{sz}$) or discharge ($Q_{sz}$). $Q_{sz}$ is zero as long as storage is below the capacity of the source zone storage. If the latter is reached, discharge from the source zone is partitioned into recharge to the transport zone ($R_{tz}$) and direct overland flow to the catchment outlet ($OF$). Overland flow is assumed to occur when the infiltration rate exceeds the infiltration capacity. This infiltration capacity is specified by a normal distribution (with mean infiltration capacity $\mu_{OF}$ and standard deviation $\sigma_{OF}$ as model parameters) to reflect spatial heterogeneity of infiltration processes. This is necessary as overland flow was observed to occur under both large and much smaller rainfall events, which would not be adequately captured by a single infiltration capacity. Output fluxes from the transport zone are evapotranspiration ($ET_{tz}$) and discharge to the catchment outlet ($Q_{tz}$); $Q_{tz}$ was assumed to be a function of storage solely ($ST_{tz}$; cf. Kirchner, 2009). Vegetation effects were not modelled. Discharge at the catchment outlet was simulated on a daily time step. The detailed equations of storage and fluxes for the hydrological model are given in Table S5 in the SM.

Our model uses the transport formulation by travel time distributions, which characterize flow dynamics within a reservoir by giving the probability density function of the time that a water parcel spends inside the reservoir before leaving *via* Q or ET, respectively (Botter et al., 2010; Hrachowitz et al., 2016; van der Velde et al., 2012). Travel time distributions also allow for the calculation of solute concentrations by convolution of travel time distributions with a relation between travel times and concentrations (Benettin et al., 2013; Botter et al., 2010). The shape of travel time distributions depends on the assumed storage selection scheme (SAS functions), which specifies the time-variance of travel times, and the preference of Q and ET to remove water of a certain age from storage (Rinaldo et al., 2015; van der Velde et al., 2012). In this study, we opted for a SAS function describing variable flow with time-varying storage selection. This means that travel time distributions are time-variant and different for ET, Q, and storage, and that the preference of discharge (Q) for water of a certain age depends on storage (i.e., SAS-function of discharge changes with storage in the transport zone; van der Velde et al., 2015; Harman, 2015). Travel time distributions were calculated for the modelled fluxes from the source and transport zone, which, in turn, yielded pesticide concentrations in $Q_{sz}$, $Q_{tz}$, and $ET_{tz}$ (i.e., C$_{sz}$, C$_{tz}$, C$_{ET}$; Table S6). A detailed description of travel time





distributions and related mixing schemes can be found in, e.g., van der Velde et al. (2012), Botter et al. (2010), and Harman (2015).

## 2.6 Pesticide model: mass transfer and transport

The applied pesticide enters the model system via the source zone (mass flux $\Phi_{inp}$ in Fig. 2) and is assumed to be initially present in the sorbed phase of the source zone, given the physico-chemical properties of S-metolachlor. The reasoning behind this is that farmers tend to apply pesticides during dry periods, after which most of the applied water will evaporate, leaving the pesticides sorbed to soil particles or in the more tightly bound soil water. In support of this assumption, previous studies have found strong sorption of S-metolachlor to surface soils, where the organic matter content is typically higher than at greater depth (e.g. Bedmar et al., 2011; Rice et al., 2002; Si et al., 2009). Subsequently, infiltration of precipitation leads to pesticide desorption and input into the dissolved phase of the source zone reservoir. Infiltration mobilizes only a fraction of the adsorbed pesticide in the source zone reservoir depending on the contact time between water and soil. Hence, pesticide concentrations in the source zone reservoir decrease with increasing water flow and thus decreasing contact time (i.e., preferential flow; see Table S6, term $(1 - e^{-lT_{sz}})$ in the equation for $C_{sz}(t)$). Pesticide in the dissolved phase is exported from the source zone via discharge ($\Phi_{sz}$), which leads to pesticide input into the transport zone ($\Phi_r$) and potentially direct transport to the catchment outlet via overland flow (i.e., in the dissolved phase after desorption; $\Phi_{of}$). The model also accounts for direct pesticide transport from the source zone to the catchment outlet in the particulate phase via eroded material ($\Phi_{er}$; without desorption). The eroded pesticide amount was assumed proportional to discharge via overland flow and stored pesticide mass in the source zone (related by model parameter $f_{er}$, Table S6). Hence, the erosion pathway ($\Phi_{er}$) removes sorbed pesticide from the source zone and thus plays an important role for the overall pesticide mass balance.

Pesticide in the transport zone can be discharged to the catchment outlet ($\Phi_{tz}$) or return to the source zone *via* evaporation from the transport zone ($\Phi_{et}$). $\Phi_{et}$ was assumed to redirect a fraction of the pesticide mass back into source zone storage ($\Phi_{ex}$; Benettin et al., 2013; Bertuzzo et al., 2013; Queloz et al., 2015) to account for incomplete uptake of pesticide in ET water by plants and pesticide release to the soil after plant uptake (Al-Khatib et al., 2002; Henderson et al., 2007). Dissolved pesticide concentrations at the catchment outlet were calculated from concentrations in overland flow ($\Phi_{of}$) and discharge from the transport zone ($\Phi_{tz}$). Concentrations and $\delta^{13}C$-values of S-metolachlor were simulated in the dissolved phase at the catchment outlet (C and $\delta^{13}C$ in Fig. 2, respectively) on a daily time step. Following Bertuzzo et al. (2013), sorption in the transport zone was not considered in the model in order to limit model complexity, and in view of the rapid decrease in the soil organic matter content with depth (i.e., from about 2.5 % at the surface to 0.6 % below 1 m depth). Table S6 in the SM shows the detailed equations for pesticide storage, mass fluxes and concentrations.

## 2.7 Pesticide model: degradation and isotope fractionation

We considered biodegradation as the main process of S-metolachlor mass reduction in the catchment (Accinelli et al., 2001; Miller et al., 1997). We simulated first-order kinetics with a constant half-life in the source zone as previously described in





Bertuzzo et al. (2013) and Queloz et al. (2015). For the transport zone, we simulated an exponential decline of the degradation rate constant with travel time (see Table S6, term $e^{-\frac{r_0}{k}(1-e^{-kT_{tz}})}$ in the equations for $C_{tz}(t)$ and $C_{ET}(t)$), which resembles a linear decrease in the degradation rate with depth given an exponential increase in travel time with depth (van der Velde et al., 2010). This mirrors slower pesticide degradation in deeper soil layers compared to the topsoil due to

decreasing microbial activity (Rodríguez-Cruz et al., 2006; Si et al., 2009).

The model was applied to explicitly calculate concentrations of light and heavy carbon isotopes contained in the pesticide. This allowed for the simulation of degradation-induced isotope fractionation: the light isotopes degrade with a rate constant $r_0^{12}$, which is related to the rate constant of the heavy isotopes ($r_0^{13}$) by the isotope fractionation factor $\alpha$ ($\alpha < 1$) as $r_0^{13} = \alpha \cdot r_0^{12}$. Modelled $\delta^{13}$C-values at the catchment outlet were calculated from the simulated concentrations of the light and heavy

carbon isotopes following Eq. (1). For model simplicity and because of the lack of more detailed information, we assumed a single degradation mechanism with one unique fractionation factor ($\alpha$). In the following, we will refer to the carbon isotopic enrichment factor $\varepsilon_C = 1 - \alpha$ (reported in ‰) instead of $\alpha$.

Considering the physico-chemical properties of S-metolachlor (Table S1), we disregarded abiotic degradation processes (e.g., photolysis) and potential associated isotope fractionation effects in the modelling. We neither simulated sorption-

induced isotope fractionation, which is considered insignificant during limited numbers of sorption-desorption steps (Kopinke et al., 2017). Moreover, we disregarded pesticide transfer to and from the atmosphere (i.e., volatilization and deposition), as we consider volatilization to be minor based on the relatively low Henry's Law constant of S-metolachlor and previous research on volatilization of S-metolachlor (Parochetti, 1978; Rice et al., 2002; Rivard, 2003). We assumed instead that the water in the pesticide spray is rapidly lost by evaporation, leaving the remaining pesticide adsorbed to soil.

**2.8 Input data and calibration**

The model simulated discharge, pesticide concentrations and $\delta^{13}$C-values from 1 September 2004 to 31 December 2012 to ensure sufficient model spin-up time. It was run with daily data for precipitation and potential evapotranspiration ($ET_{pot}$) from the meteorological station Waltenheim sur Zorn. Pesticide input rates and dates of pesticide application for each simulation year were set to the application rates and dates of S-metolachlor in 2012 (obtained from the survey among the

farmers). No pesticide was present in the model domain at the beginning of the simulation.

Calibration was performed for 18 model parameters (ranges of parameter values are provided in Table S7) against the following measured data: daily average of discharge at the catchment outlet (Fig. 1) between 9 March and 14 August 2012; 33 flow-proportional samples of S-metolachlor concentrations at the catchment outlet between 20 March and 21 August 2012, among which six with $\delta^{13}$C-values; and one grab sample of S-metolachlor concentrations at the catchment outlet on 20

November 2012. The initial $\delta^{13}$C-value of the applied S-metolachlor was specified a priori as $\delta^{13}C_0$ = -32.5 ‰ to be able to simulate $\delta^{13}$C-values as low as -32.4 ‰ measured at the catchment outlet. The $\delta^{13}C_0$-value in the model was thus more negative than the $\delta^{13}$C-value of the S-metolachlor formulation exclusively used in the plot experiment ($\delta^{13}$C = -31.9±0.3 ‰).





We did not include spatial variability in the initial $\delta^{13}$C-value of S-metolachlor due to the predominant use (about 80 %) of one commercial S-metolachlor product according to the survey among the farmers.

Model performance of each parameter set was evaluated with respect to the Nash-Sutcliffe coefficients for discharge ($NS_Q$), concentrations ($NS_C$), and $\delta^{13}$C-values of S-metolachlor ($NS_{\delta13C}$; see section S5 in the SM for a detailed description). First,

preliminary model calibrations were run to determine the parameter set with the best possible model fit in terms of $NS_Q$, $NS_C$, and $NS_{\delta13C}$. In view of the relatively few field data, a range of values for the $NS$-efficiency was defined around the best fit that indicates equally suitable (i.e., behavioural) model results similar to the GLUE approach (Beven, 2012). Finally, 10,000 behavioural parameter sets were determined by optimizing model parameters in 10,000 additional calibration runs until reaching behavioural NS-efficiencies.

## 3 Results and Discussion

### 3.1 Monitoring results: pesticide concentrations and $\delta^{13}$C-values

Concentrations of dissolved S-metolachlor in surface runoff from the plot between April and mid-July were highest during the first runoff event (64.1 µg L$^{-1}$ on 17 April; Table S2) that followed S-metolachlor application (12 April; Fig. 3a). Concentrations at the plot remained above 10 µg L$^{-1}$ during the study period (except for 19 June), which underscores the

persistence of S-metolachlor in the soil throughout summer (field-derived half-life of 54 days; Lefrancq, 2014). At catchment scale, S-metolachlor concentrations were below 0.1 µg L$^{-1}$ in March and April (Fig. 3b and Table S4), which indicates negligible pesticide residues from previous years. S-metolachlor was mainly mobilized during the extreme rainfall-runoff event on 21 May with maximum concentrations of 62.1 µg L$^{-1}$ (dissolved phase) as opposed to low concentrations before. During this runoff event, concentrations at the catchment outlet were similar to concentrations at the plot on 17 April

(Fig. 3a and b). In subsequent runoff events, concentrations at the catchment outlet gradually decreased from 6.1 µg L$^{-1}$ (end of May) to around 0.1 µg L$^{-1}$ (August). As the plot samples were exclusively fed by surface runoff, larger concentrations at the plot compared to the catchment highlight the importance of surface runoff as transport route for S-metolachlor in the Alteckendorf catchment. At catchment scale, such localised surface runoff re-infiltrated before reaching the catchment outlet in the first weeks of the study period. The first occurrence of surface runoff at the catchment outlet was thus later (i.e., on 2

May), after a sealing crust on the silty soil had developed, which resulted in large areas of surface runoff close to the catchment outlet. In addition to limited pesticide application before May, this explains the lack of concentration peaks in April, as opposed to what has been observed at the plot.

Concentrations in the grab samples from the drain outlet reached a maximum of 2.21 µg L$^{-1}$ following the extreme rainfall event; the mean value of all samples was 0.28±0.52 µg L$^{-1}$ (n = 16; Fig. 3c and Table S3). However, potential concentration

peaks during high flow could not be captured, as the drain outlet was below the water level and thus not accessible under high-flow conditions. Nonetheless, assuming minor concentration variations outside high-flow periods, the low





concentrations at the drain outlet suggest a secondary contribution to herbicide export from drain outflow and groundwater seepage.

The decreasing concentrations at the plot and catchment outlet with successive runoff events in May and June indicate a gradual depletion of the topsoil herbicide pool, which might be ascribed to herbicide transport via surface runoff, infiltration into deeper soil layers, and/or degradation between the runoff events. The occurrence of degradation is supported by the field CSIA data: $\delta^{13}$C-values of S-metolachlor at the plot and catchment were in the range of the applied product (-31.9±0.31 ‰ in the application tank) during the first runoff events in May (-32.2 to -31.6 ‰ at the plot and -32.4 to -31.6 ‰ at the catchment outlet) and became gradually enriched in June and July, yielding an increase by 2.6 ‰ at the plot (between 22 May and 10 July; Fig. 3a and Table S2) and by 2.5 ‰ at the catchment outlet (between 21 May and 17 July; Fig. 3b and Table S4). As opposed to the concentration data, CSIA data thus give clear evidence of in situ degradation of S-metolachlor. The magnitude of isotopic enrichment indicates a similar extent of degradation at the plot and catchment scale. This suggests that degradation-induced fractionation effects primarily occurred in the topsoil, as the plot experiment only captured herbicide in surface runoff remobilized from the topsoil. Given that pesticide residues from previous years in the topsoil were secondary, this also implies that the isotopic enrichment occurred in the course of around two months.

As the first samples do not show significant enrichment (assuming the same initial $\delta^{13}$C-value for the bulk pesticide as for the pesticide applied at the plot), degradation might have mainly occurred from June on, following a period of little degradation in April and May. This might result from the generally limited availability of sorbed pesticides for microbial degradation (e.g., Dyson et al., 2002; Guo et al., 2000; Park et al., 2003), which, in turn, hampers degradation-induced fractionation before the first rainfall event of May 21. Moreover, little degradation before June may be explained by lower soil temperature in spring (daily mean air temperatures mostly between 5 and 10°C in April and around 10°C on several days in May), resulting in lower microbial activity (Dinelli et al., 2000; Barra Caracciolo et al., 2005).

Acetochlor concentrations at the plot peaked during the first runoff event on 17 April (Fig. 3d), but at a much lower concentration than S-metolachlor (1.8 µg L$^{-1}$), as acetochlor had not been applied at the plot in 2012. Its frequent detection in the plot samples must, therefore, be ascribed to contamination from surrounding fields due to drift, or, possibly, to applications in previous years. Acetochlor at the plot shows a pronounced isotopic enrichment between two samples from the end of May and mid-June, respectively (Fig. 3d). However, as it had not been applied at the plot, other processes than transformation might be at the origin of this enrichment.

Concentrations of acetochlor at the catchment outlet (Fig. 3e and Table S4) were comparable to those of S-metolachlor (Fig. 3b). As observed for S-metolachlor, concentrations of acetochlor at the catchment outlet were highest in response to the rainfall event on 21 May, and rapidly decreased afterwards. However, the last two samples show a concentration increase following very low concentrations in July. This could result from a second application of the compound (not suggested by the survey among the farmers), or a delayed arrival of the pesticide in the drainage network due to slow pesticide transport through the soil matrix. In contrast, concentrations of acetochlor at the drain outlet were below 1 µg L$^{-1}$ in all samples (Fig.





3f and Table S3). Hence, this supports the assumption of a minor contribution of drain outflow and groundwater seepage to overall pesticide export.

Similar to S-metolachlor, $\delta^{13}$C-values of acetochlor at the plot and catchment scale became significantly enriched between May and July (enrichment above 3 ‰; Fig. 3d and e, and Table S2 and S4). This confirms the hypothesis of similar isotope fractionation effects at both spatial scales and the dominant role of degradation in the topsoil compared to deeper soil layers. However, in the case of acetochlor, plot samples show a systematic shift of approximately 4 ‰ towards more depleted values compared to the catchment samples. As acetochlor was not applied at the experimental plot, it is unclear whether this shift reflects isotope fractionation effects due to, e.g., volatilization, or deposition of a more depleted acetochlor formulation used on another field.

## 3.2 Modelling results: multiple model calibrations

The model generally captured the measured discharge at the catchment outlet (Fig. 4b). It also succeeded in reproducing the low S-metolachlor concentrations prior to pesticide application and the concentration peaks during high-flow conditions, with the maximum occurring in response to the extreme rainfall event on May 21 (Fig. 4c). The ranges of modelled concentrations and $\delta^{13}$C-values were comparably narrow between May and August; they were much wider outside this period when model results were not constrained by calibration to samples with CSIA data and concentrations above the detection limit, respectively (Fig. 4c and d).

Although erosion and overland flow in the model were triggered several times during the study period, their impact on concentrations was minor apart from the response to the extreme rainfall event. Hence, the model underestimated peaks in discharge and concentrations (Fig. 4b and c). Some calibrations yielded concentrations in the low ng L$^{-1}$ range in late summer due to the absence of pesticide release from the source zone (Fig. 4c). In early autumn, however, pesticide concentrations increased following precipitation and associated pesticide release. Accordingly, the model simulated persistence of S-metolachlor in topsoil throughout summer, which is in agreement with the detection of S-metolachlor at the catchment outlet even several months after the pesticide applications (e.g., concentration of 0.1 µg L$^{-1}$ in the grab sample on 20 November 2012).

The model predicted a gradual increase in $\delta^{13}$C-values after pesticide application in April and May. This reflects isotopic enrichment in discharge from the transport zone due to pesticide degradation. Simulated $\delta^{13}$C-values outside the study period were associated with a wide range of possible values, and highlight a pronounced isotopic enrichment in early autumn (Fig. 4d). Moreover, simulated $\delta^{13}$C-values suggest the occurrence of local minima following rainfall events. Rainfall events mobilized pools of sorbed and thus non-degraded pesticide in the source zone, which was thereby transferred to the transport zone (and secondarily to overland flow). The temporal predominance of non-degraded pesticide pools in discharge led, in turn, to relatively low $\delta^{13}$C-values at the catchment outlet. In addition, rainfall events resulted in temporally decreasing travel times, which limited reaction time and thus isotope fractionation in the transport zone. The minima in $\delta^{13}$C-values were most distinct after the second S-metolachlor application at the plot (May 1), and after the extreme rainfall event on May 21 (Fig.



4d); they were much less pronounced in late autumn and winter, when most of the pesticide had already been removed from the source zone. Due to the limited temporal resolution of the field CSIA data, it is not possible to conclude whether these fluctuations occurred in reality. However, because of these model results, it becomes apparent that CSIA measurements both before and after rainfall events are particularly valuable to understand the extent of remobilization of sorbed pesticides from the source zone.

Before the first minimum in $\delta^{13}$C-values on May 1, the increase in $\delta^{13}$C-values levelled off in most calibration runs, attaining an upper limit of isotopic enrichment. The pesticide reaching the catchment outlet before May is associated with long travel times, which results in a significant decrease in the rate of pesticide degradation in the transport zone (exponential decrease with travel time) and thus inhibits further isotope fractionation. As with the minima in $\delta^{13}$C-values following rainfall events, it would be beneficial to compare these model results with field CSIA data. However, as CSIA cannot currently be performed at such low pesticide concentrations without interferences, measurements during these time periods might, for now, not provide additional information on pesticide fate.

The calibration results yielded a smaller half-life in the source zone (mean = 5.6 d; range from 5.1 d to 14.7 d) than reported half-lives for S-metolachlor (15 - 54 d; Table S1). In contrast, half-lives in the transport zone were considerably larger due to decelerated degradation with longer travel times (i.e., increasing depth). For example, averaged over all simulations, model half-lives in the transport zone were 63.9 d and 2310.2 d at depths corresponding to travel times of six months and one year, respectively. This yielded the simulated long tailing of S-metolachlor concentrations in discharge from the transport zone (Fig. 4c).

### 3.3 Modelling results: quantification of pesticide transport and degradation

The model allows tracking pesticide transport and degradation in all compartments of the simulated system (Table 1). The mean pesticide export from the catchment based on 10,000 simulations was 4.6±5.3 % of the applied mass in the study period and 4.7±5.3 % in the entire year 2012, of which the majority occurred via erosion (i.e., 3.8±5.2 % of the total mass in both time frames). The minor difference in pesticide export between the two time frames underlines the importance of the extreme rainfall event, which entailed more than 99 % of the overall pesticide transport via erosion in 2012. Discharge from the transport zone accounted for the remaining 0.3 % and 0.4 % in the study period and 2012, respectively, which is in line with low concentrations at the drain outlet (Fig. 3c). The average extent of pesticide degradation in 2012 was 92.6±5.9 % of the applied mass (80.8±6.5 % in the source zone and 11.8±5.0 % in the transport zone, respectively).

The differing mass balance terms for pesticide transport between 2011 and 2012 highlight the importance of erosion for simulated pesticide export from the catchment. Erosion accounted for the bulk of pesticide transport in 2012, whereas it barely occurred in the simulation of 2011 (Table 1). Similarly, pesticide export via the transport zone in 2011 was minor with a proportion of 0.1 % of the total applied pesticide mass. Therefore, pesticide retention in the catchment was larger at the end of 2011 (4.4±2.5 %) compared to 2012 (2.7±2.0 %).





### 3.4 Insights on pesticide fate and transport from the model

The model allowed for testing different representations of pesticide transport and degradation. We set up three modified models and calibrated each in 1000 simulations against the same data as the original model. First, 1000 simulations were run without pesticide degradation and calibrated against measured discharge and concentrations. Concentrations remained above

10 µg L$^{-1}$ during most of the year and the observed concentration decline from May to September could not be reproduced even with increased erosion (Fig. S1). Hence, this illustrates the need for simulating pesticide degradation in the model. Second, the original model was applied without pesticide transport via erosion (but overland flow was kept) and calibrated against discharge, concentrations and δ$^{13}$C-values. The lack of mass loss via erosion was counterbalanced by increased degradation in the transport zone. In contrast, degradation in the source zone decreased, on average, relative to the original

model, which resulted in comparable isotopic enrichment factors with the two model setups (i.e., mean of $\varepsilon_C$ = -1.0±0.4 ‰ vs. $\varepsilon_C$ = -0.9±0.3 ‰ in the original model). Overall, the simulation results of the model without erosion are similar to those of the original model (Fig. 5a). However, the behavioural simulations of the model with erosion (i.e., original model) yielded both better model fits (i.e., mean $NS_C$ of 0.83 vs. 0.73 for concentration, Eq. (S5); and mean $NS_{\delta13C}$ of 0.94 vs. 0.93 for δ$^{13}$C, Eq. (S6)) and a smaller range of concentrations and δ$^{13}$C-values outside the calibration period than the model without

erosion. This implies that adding erosion improves the model representation of pesticide transport.

The third modification consisted in a model with erosion and degradation, with the latter being implemented by a constant degradation half-life in both the source and transport zone instead of a declining first-order degradation rate constant with increasing travel time (cf. equation for $C_{tz}(t)$ in Table S6). This model was again calibrated against measured discharge, concentrations, and δ$^{13}$C-values. Differences between the original and the modified model with a constant degradation half-

life (Fig. 5b) were even more distinct than for the model without erosion (Fig. 5a). Again, both improved model fits (i.e., $NS_C$ of 0.83 vs. 0.74 with a constant degradation half-life; $NS_{\delta13C}$ did not change) and much smaller simulated ranges of concentrations and δ$^{13}$C-values outside the study period demonstrate the improved representation of the transport process with the model that includes a declining degradation rate constant with increasing travel time. This is due to degradation in the transport zone being faster in the model with a constant degradation half-life compared to the original model, which led

to substantially lower concentrations than the measured values (note the log-scaling in (Fig. 5b, iii)) and more enriched δ$^{13}$C-values in autumn and spring (i.e., when the majority of the pesticide is in the transport zone) compared to the original model. The extent of degradation in the source zone was, on the contrary, smaller than in the original model, which resulted in slightly less overall degradation in the model with a constant degradation half-life (i.e., mean of 90.8 % vs. 92.6 % in the original model). Consequently, with a mean of $\varepsilon_C$ = -1.3±0.7 ‰, the calibrated enrichment factor was slightly larger than

with the original model (i.e., $\varepsilon_C$ = -0.9±0.3 ‰). Whereas conceptual landscape models generally assume a constant degradation half-life for pesticides over the entire subsurface depth (e.g., SWAT model, Nietsch et al., 2011; ZIN-AgriTra model, Gassmann et al., 2013; IMPT model, Pullan et al., 2016; REXPO model, Wittmer et al., 2016), these results support





the simulation of a decreasing degradation rate constant with increasing subsurface depth. This can be easily implemented in our model by means of travel-time distributions.

The model results suggest that persistence of S-metolachlor in the environment is strongly coupled to pesticide sorption in the source zone. By assuming no decay of sorbed pesticides in the source zone and desorption of pesticides during rainfall

events, we were able to describe the observed long persistence and relatively small increase in $\delta^{13}$C-values with time. In our conceptual model, each rainfall event mobilizes a part of the non-decayed sorbed pesticide, which effectively lowers the $\delta^{13}$C-value of the mobile soil water after each rainfall event. The pesticide concentration of pulses with low $\delta^{13}$C-value decline with increasing time after application as most of the sorbed pesticides has been transported out of the source zone (cf. section 3.2). Based on these results, but out of reach for our study, further investigations are needed to check for the

occurrence of such episodes of declining $\delta^{13}$C-values after rainfall events long after pesticide application in order to support our model concepts.

In addition to testing of different model setups, we used our conceptual model to assess a field-integrated enrichment factor ($\varepsilon_C$). We propose that travel-time distribution modelling inherently accounts for dispersion and flowpath mixing (Hrachowitz et al., 2016) and is thus able to mirror attenuation of potentially large isotope fractionation due to dispersion and mixing in

open systems (Abe and Hunkeler, 2006; Lutz et al., 2013; van Breukelen and Prommer, 2008). In contrast, enrichment factors derived from field CSIA data via the Rayleigh equation (Eq. S1) are prone to underestimate the "true" enrichment factor in open systems (Abe and Hunkeler, 2006; van Breukelen, 2007). Hence, conceptual travel-time based models represent an alternative way of assessing field-integrated enrichment factors, which might improve in accuracy with increased temporal resolution of CSIA samples. With a mean $\varepsilon_C$ of -0.9±0.3 ‰ and a best-fit value of -1.13 ‰ in this study,

the calibrated $\varepsilon_C$ for S-metolachlor is smaller (i.e., corresponding to less fractionation) than experimentally determined $\varepsilon_C$-values for the chloroacetanilides alachlor ($\varepsilon_C$ = -2.0±0.3 ‰) and acetochlor ($\varepsilon_C$ = -3.4±0.5 ‰; Elsayed et al., 2014). This is also the case for the two alternative model setups discussed above (i.e., mean values of $\varepsilon_C$ = -1.3±0.7 ‰ with a constant degradation half-life and $\varepsilon_C$ = -1.0±0.4 ‰ without erosion, respectively). However, enrichment factors may differ between compounds of the same pesticide group and even between microbial degradation pathways of the same compound

(Hartenbach et al., 2008; Meyer and Elsner, 2013; Meyer et al., 2009; Penning et al., 2010; Penning and Elsner, 2007).

## 3.5 Degradation assessment with the Rayleigh equation and the model

Previous modelling studies have demonstrated that the Rayleigh equation (Eq. S1) systematically underestimates the actual extent of degradation (Abe and Hunkeler, 2006; van Breukelen and Prommer, 2008; van Breukelen and Rolle, 2012; Lutz et al., 2013). The underestimation by the Rayleigh equation results from attenuation of effective isotope fractionation by

dispersion and mixing processes in real-world flow systems, which is disregarded in the Rayleigh equation. In this study, we compared the extent of degradation known from the simulated concentrations and the model mass-balance, respectively, with the one estimated by the Rayleigh equation. This enables us to assess the reliability of the Rayleigh equation to estimate





degradation at catchment scale. We made use of simulated (virtual) CSIA data to assure unbiased comparison of continuous time-series.

Assuming the same enrichment factor and initial isotopic signature of the applied product as in the best-fit simulation (i.e., $\varepsilon_C$ = -1.13 ‰ and $\delta^{13}C_0$ = -32.5 ‰, respectively), the Rayleigh equation approach gave an extent of degradation of $ED_{Rayleigh}$ =

94.1 % (Eq. S2) for the simulated $\delta^{13}C$-value of -29.4 ‰ on 17 July (i.e., best-fit simulation on last day of field CSIA data). In comparison, based on the simulated concentrations of S-metolachlor and a conservative tracer at the catchment outlet (for the entire simulation period from 2004 to 2012), the actual extent of degradation on 17 July was $ED_{Sample}$ = 99.9 % (Eq. S3) in the best-fit simulation. Overall, the Rayleigh equation applied to (virtual) CSIA data from the catchment outlet underestimated the actual extent of degradation as occurred for the outlet sample (i.e., $ED_{Rayleigh} < ED_{Sample}$; Fig. 6e) in

agreement with earlier studies (see above). In this study, the underestimation of $ED_{Sample}$ was more pronounced during high flow periods (soon after pesticide application) compared to baseflow periods (Fig. 6), which is in agreement with Lutz et al. (2013). High flow periods in the model were associated with input of barely degraded pesticide, which, therefore, masked significant isotope fractionation in pesticide from the transport zone.

Employing the best-fit estimate of $\varepsilon_C$ = -1.13 ‰, $ED_{Rayleigh}$ was always larger than the actual extent of degradation in the

entire catchment derived from the mass balance of the model in 2012 ($ED_{Catchment}$; Fig. 6e). For example, $ED_{Catchment}$ was 72.9 % on 17 July, which is considerably smaller than the Rayleigh estimate for the outlet sample ($ED_{Rayleigh}$ = 94.1 %; both for the best-fit simulation). In contrast to $ED_{Rayleigh}$ versus $ED_{Sample}$, the deviation between $ED_{Rayleigh}$ and $ED_{Catchment}$ was largest during baseflow periods (with relative deviations of more than 50 % during the study period), and sharply decreased for high flow conditions (Fig. 6). This can be explained by the large extent of degradation for pesticide in slow discharge

from the transport zone during baseflow, which is not representative of the overall extent of degradation in the catchment, as opposed to discharge of less-degraded pesticide during high flow conditions (associated with recent desorption and short travel times). Accordingly, as most of the pesticide was contained in the source zone reservoir especially shortly after pesticide application, $ED_{Rayleigh}$ was closer to $ED_{Catchment}$ for samples with a dominant imprint of the source zone (i.e., during high flows). This suggests that CSIA-based degradation estimates are most representative of catchment-scale decay for

samples taken during high flow conditions, when pesticide discharge occurs via shallow soils mainly. This result further emphasizes the need for studies on topsoil sorption and desorption of S-metolachlor under field conditions and realistic application scenarios.

Laboratory-derived carbon enrichment factors of pesticides are subject to uncertainty ranges typically between ±0.1 ‰ and ±1.0 ‰ (Elsayed et al., 2014; Meyer and Elsner, 2013; Wu et al., 2014). In order to assess the uncertainty in Rayleigh

equation estimates, we considered an uncertainty range of ±0.5 ‰ for $\varepsilon_C$ around the best-fit value (i.e., -1.63 ‰ ≤ $\varepsilon_C$ ≤ -0.63 ‰) and assumed that the simulated $\delta^{13}C$-values of the best-fit calibration describe the actual $\delta^{13}C$-values. The Rayleigh equation yielded the largest uncertainty range in degradation estimates in response to rainfall events (green area in Fig. 6e). For example, whereas the best-fit estimate with $\varepsilon_C$ = -1.13 ‰ gave $ED_{Rayleigh}$ = 94.1 % on July 17, $ED_{Rayleigh}$ ranged between 85.9 and 99.4 % on that day with $\varepsilon_C$ = -1.13±0.5 ‰. Nonetheless, the mean absolute deviation from the best-fit simulation




was below 3.5 % in 2012 (maximum deviation of 21.1%). Hence, despite an uncertainty range of ±0.5 ‰ in $\varepsilon_C$, the Rayleigh equation estimate remained within an acceptable error band and thus allowed for a first (conservative) approximation of the extent of degradation at the catchment outlet.

Overall, these results suggest that CSIA data and CSIA-based (i.e., Rayleigh-equation based) estimates are not representative for the entire catchment. Hence, the modelling approach is required in order to upscale CSIA information on degradation to the catchment scale, which is crucial in the quantification of pesticide residues in the catchment.

# 4 Conclusions

This study presents the first measurements of herbicide CSIA-data at catchment scale. Carbon isotope ratios of the herbicides S-metolachlor and acetochlor at the catchment outlet indicated a delayed onset of pesticide degradation after a period of little degradation during spring. S-metolachlor degradation and transport was simulated in a conceptual (i.e., parsimonious mathematical) model based on travel time distributions. The simulation results underlined that assessment of pesticide degradation at catchment scale with a constant half-life (e.g., derived from topsoil studies) might lead to overestimation of pesticide degradation in deeper soil layers and thus overly optimistic expectations on environmental protection. In addition, the model results demonstrated that degradation estimates via the Rayleigh equation are considerably larger than the overall extent of degradation within the catchment (calculated from the model mass-balance). This implies that a large isotopic enrichment of streamwater samples does not necessarily correspond to a large extent of pesticide degradation in the entire catchment, as it can also result from transient contributions of flowpaths associated with more pronounced degradation relative to the bulk of the pesticide mass. Moreover, laboratory-derived enrichment factors might not be suitable for degradation assessment with the Rayleigh equation at catchment scale, provided that such values are available for the examined compound at all. Hence, conceptual modelling based on travel time distributions can prove beneficial as complementary approach in the evaluation of pesticide degradation for streamwater samples and the entire catchment.

The model indicated that the dynamics of CSIA data at the catchment outlet were highly responsive to changing hydrological conditions (e.g., following rainfall events). As CSIA measurements in this study were only possible after rainfall-events and not during low-flow periods, we were not able to corroborate this model result. This implies that additional CSIA data at higher temporal resolution (ideally at the same resolution as concentration data) and at later times of the study period especially during low flow conditions can yield a much clearer picture of pesticide transport and degradation. Such data might further improve the representation of pesticide degradation in models building on the conceptual model presented in this paper, and allow for the use of CSIA data as tracer for different flowpaths at catchment scale.

Depending on the study-compound properties, future modelling studies might include additional processes such as re-equilibration between pesticide in the dissolved and particulate phase. Similarly, incorporation of sorption processes in the transport zone would be an important future step with this model setup, as sorption might still occur in deeper soil despite decreasing organic matter content with depth. Nonetheless, insights from this study highlight how conceptual modelling of



pesticide degradation and transport with travel-time distributions can advance our understanding of pesticide fate and contribution of different transport pathways at catchment scale, and pinpoint knowledge gaps for which additional measurements are required, especially when applied in a combined experimental and modelling approach. As travel-time based models are computationally non-intensive and can be applied in the absence of spatially distributed information on

parameters (e.g., soil hydraulic properties), they can also be useful for the implementation of monitoring and regulation tests in agricultural catchments.

## Data availability

The measured data can be found in the supplement to this article (section S2).

## Author information

The field experiment was designed and performed by G. Imfeld, M. Lefrancq and S. Payraudeau. M. Lefrancq performed the concentration analyses. O. Elsayed performed the isotopic analyses. B. van Breukelen, S. Lutz and Y. van der Velde designed the model study. Y. van der Velde and S. Lutz developed the model code. S. Lutz performed the simulations and prepared the manuscript with contributions from all co-authors.

## Competing interests

The authors declare that they have no conflict of interest.

## Acknowledgements

This research has been financially supported by the European Union under the 7th Framework Programme (project acronym CSI:ENVIRONMENT, contract number PITN-GA-2010-264329; and Grant agreement no. 603629-ENV-2013-6.2.1-Globaqua), and the PhytoRET project (C.21) of the European INTERREG IV program Upper Rhine. Marie Lefrancq was

supported by a fellowship of the Rhine-Meuse Water Agency. We acknowledge the help and support of Matthias Gehre and Ursula Günther (Department of Isotope Biogeochemistry, UFZ Leipzig) for the stable isotope measurements.

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





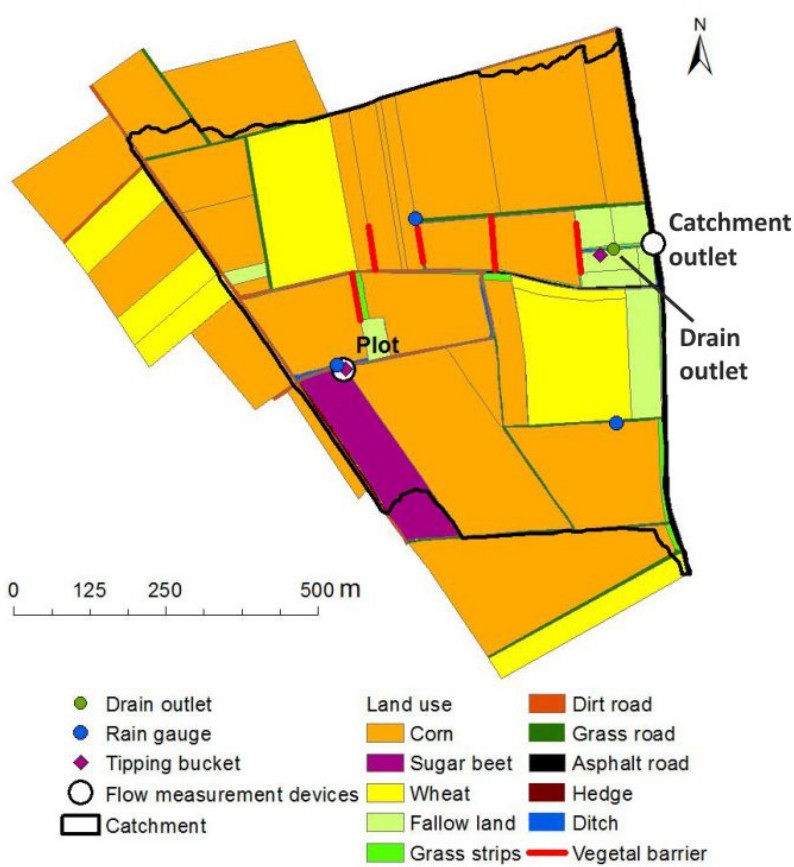

**Figure 1: Scheme of the Alteckendorf catchment (Bas-Rhin, France) with land cover and crop types. Samples were taken at the plot outlet, drain outlet and catchment outlet.**




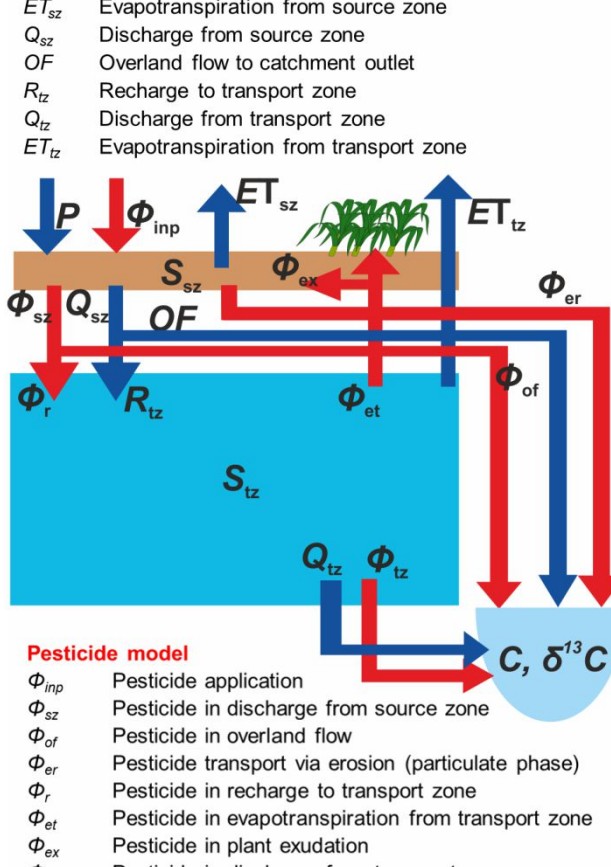

**Figure 2: Model scheme showing water flow (blue arrows) and pesticide transport routes (red arrows) from the source zone (brown box) and transport zone (blue box) to the catchment outlet (light blue semi-circle).**




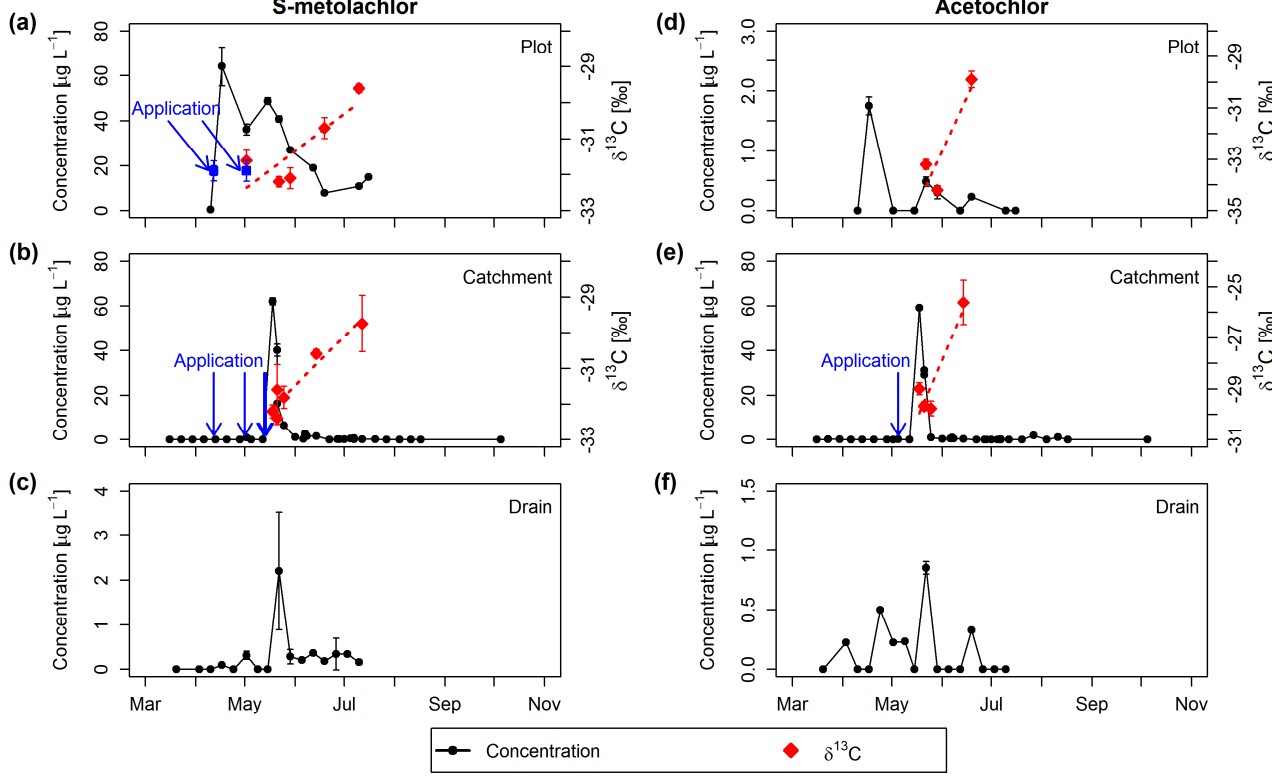

**Figure 3: Concentrations (black dots) and δ¹³C-values (red diamonds) of S-metolachlor (left panels) and acetochlor (right panels) at the plot (a, d), catchment outlet (b, e) and drain outlet (c, f). Standard deviations of replicate measurements are indicated by vertical error bars. Linear regression of δ¹³C-values is shown as dashed red lines (coefficients of determination: R² = 0.70 (a), R² = 0.92 (b), R² = 0.80 (d), and R² = 0.86 (e)). Dates of reported pesticide application in the study catchment are indicated in blue (including the δ¹³C-values of the applied pesticide in the plot experiment in panel a).**





**Figure 4: Daily precipitation (a), and measured (red lines) and modelled time series for discharge (b), S-metolachlor concentrations (c; note the log-scaling) and δ13C-values (d) at the catchment outlet in 2012. The black line indicates the results of the calibration run with the best fit in terms of the mean of NS_Q, NS_C, and NS_δ13C. Shaded areas show the range between the 5- and 95-percentiles of all simulation results. Brown bars in (a) indicate the dates of pesticide application.**





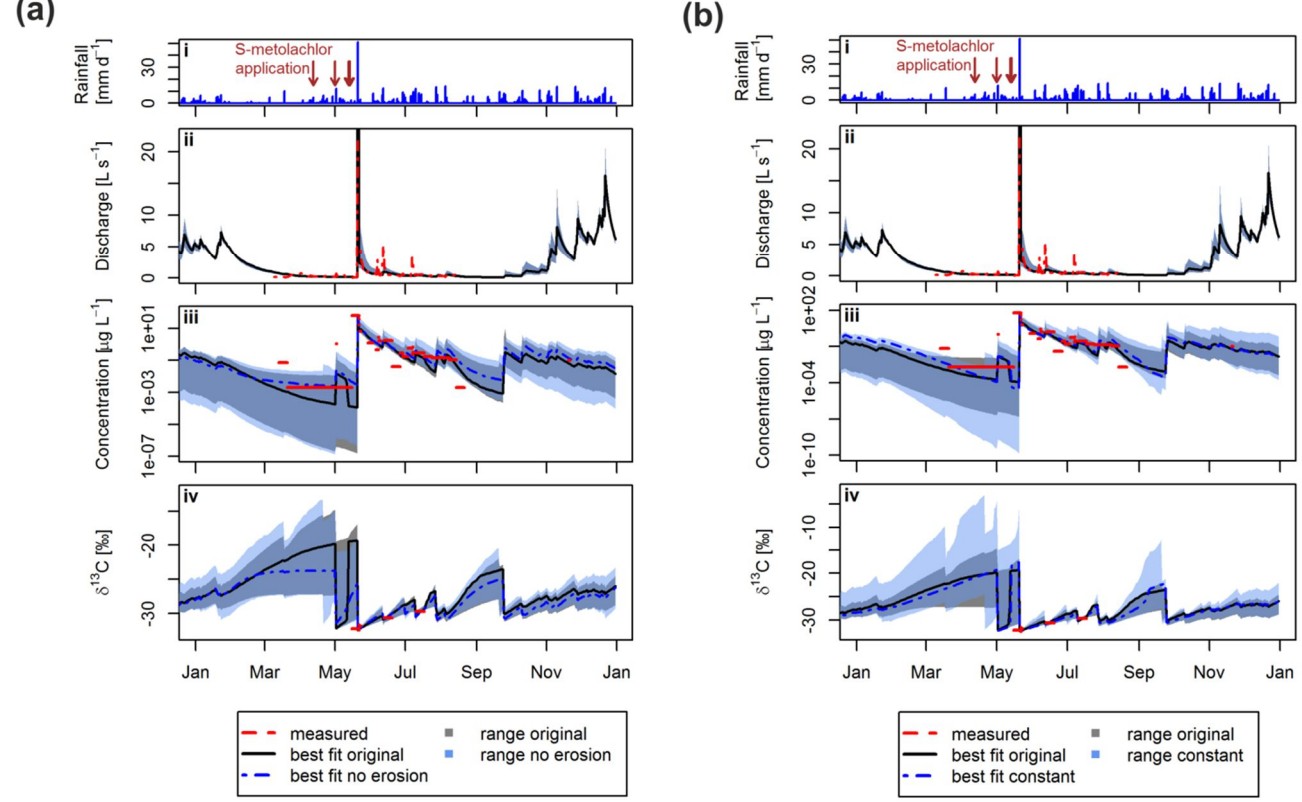

**Figure 5: Comparison between the original model and modified model setups: without erosion (a) and with a constant degradation half-life in the transport zone (b), respectively. Measured (red lines) and modelled time series are given for discharge (a, ii and b, ii), S-metolachlor concentrations (a, iii and b, iii; note the log-scaling) and $\delta^{13}$C-values (a, iv and b, iv) in 2012. Best-fit simulations for the modified and original model setups are indicated as dashed blue and solid black lines, respectively. Shaded areas show the range between the 5- and 95-percentiles of 1000 calibration runs with the modified models (transparent blue) and 10,000 calibration runs with the original model (dark grey), respectively. Blue bars in (a, i and b, i) indicate daily precipitation.**



**Figure 6: Measured (red lines) and modelled time series (black lines; best-fit simulation) for discharge (b), S-metolachlor concentrations (c; note the log-scaling) and $\delta^{13}$C-values (d), and extent of degradation as obtained by the Rayleigh equation ($ED_{\text{Rayleigh}}$, dotted green line; range of $ED_{\text{Rayleigh}}$ for $\varepsilon_C$ = -1.13±0.5 ‰, green area) and as known from the model for the sample ($ED_{\text{Sample}}$, solid brown line) and for the entire catchment ($ED_{\text{Catchment}}$, dashed purple line; e). Blue bars in (a) indicate daily precipitation.**



**Table 1: Model mass-balance for degraded and transported pesticide in 2011, 2012, and the study period (12 April to 17 July). Numbers are relative to the annually applied pesticide mass.**

|  | Study period[a] | 2012 | 2011 |
|---|---|---|---|
| **Degradation [%]** | **74.7±9.5[b]** | **92.6±-5.9** | **95.3±2.6** |
| Source zone [%] | 71.3±9.6 | 80.8±6.5 | 90.2±4.5 |
| Transport zone [%] | 3.4±1.9 | 11.8±5.0 | 5.2±2.3 |
| **Transport to catchment outlet [%]** | **4.6±5.3** | **4.7±5.3** | **0.2±0.7** |
| Overland flow [%] | 0.5±0.4 | 0.5±0.4 | 0 |
| Eroded [%] | 3.8±5.2 | 3.8±5.2 | 0.1±0.6 |
| Discharge from transport zone [%] | 0.3±0.2 | 0.4±0.2 | 0.1±0.1 |
| **Total [%]** | **79.3±8.2** | **97.3±2.0** | **95.6±2.5** |
| **Pesticide retention [%]** | **20.7±8.2** | **2.7±2.0** | **4.4±2.5** |

[a] 12 April to 17 July: from first application of S-metolachlor to last sampling day for CSIA

[b] mean ± standard deviation of all runs