# Peer review of "Pesticide fate at catchment scale: conceptual modelling of stream CSIA data"

_Hydrology and Earth System Sciences, 2017_

## Referee Comment (RC1) · Anonymous Referee #1 · 8 May 2017

This manuscript combines field measurements (pesticide concentration and isotopic composition) with modeling to investigate the fate of pesticides in the hydrological cycle. The modeling part uses a state-of-the-art approach that features description of water ages using state-dependent SAS functions.

Overall, the MS makes a significant contribution to the literature on this topic by extending current models to the analysis of compound specific stable isotopes (CSIA). Moreover, it represents a good example of how data and models should be combined to gain the maximum knowledge of the underlying processes. There are however some issues that need to be addressed before publication.

The authors decided to put most of the model technical details in the supplementary material, which is a viable option. However, the description in the main text does not

stand alone and the reader if forced to go back and forth between the text and the Supplementary Tables. So I suggest to either put an even more concise version of the methods in the text (and develop a more detailed version in the SM), or put the equation and parameter description in the main text.

I had a hard time following the equations in Table S5 because there is a mixture of continuous (differential equations) and finite difference equations. The author should decide one way to present the model and stick with it. I would suggest to use a continuous formulation. How this is then discretized into a finite difference equation for the numerical evaluation is quite trivial. For instance, $ET_{sz}$ should be

$$ET_{sz}(t) = ET_{pot}(t) \; if \; S_{sz} > 0$$

and

$$ET_{sz}(t) = 0$$

otherwise. Or, using the the Heaviside function

$$ET_{sz}(t) = ET_{pot}(t)H(S_{sz})$$

Discharge then reads:

$$Q_{sz}(t) = (P(t) - ET_{sz}(t))H(S_{sz} - S_{max})$$

and so on for the other equations. Otherwise the authors could write all difference equations, including the two storage balance equations. Also in this case there are some typos in the remaining equations. For instance in the equation for discharge from the $sz$, the symbol $d$ should not appear in the numerator. The same applies to the $ET_{tz}$ equation.

Table S5. $R_{max}$ seems like a maximum recharge rate, but please note that the description is missing from Table S7. Moreover, when $Q_{sz} > R_{max}$, where is the remaining flux going? In this equation, I was expecting to see $Q_{sz} - OF$, to account for the fraction
of $Q_{sz}$ not going to recharge the $tz$. Something is unclear in these equations, please clarify.

Table S6. $C_0(t)$ is computed assuming a well mixed reactor (i.e. total mass divided by storage). However, this seems to contradict the model formulation which assumes that every parcel of water has a certain pesticide concentration that depends on the age, and the age distribution differs from the well mixed one. The rational for this choice must be explained.

Page 9, lines 6-9. I read these lines a couple of time but I could not figure out exactly what was actually done. Which algorithm did you use for calibration. How large was the NS-efficiency? And the NS-efficiency range? Please expand and clarify on this. With 18 free parameters, the calibration is always going to be a critical point.

The authors should show the distribution of the "behavioral parameters". Were the parameters identifiable? With such a high parameter vs data ratio, I am expecting a quite broad distribution. This should help explaining why the model could be calibrated reasonably well also without degradation.

I would anticipate in the model description that some assumptions will be relaxed later, as shown in the result section. Otherwise the reader would continue reading wondering whether all the complexity is really necessary.

**MINOR COMMENTS**

Page 2. Line 22: I would move "provided that ... non-toxic" at the end of the sentence.

Page 7. Line 13. Is this type of modeling of desorption introduced here for the first time? If so, please expand a little the description. Otherwise refer to other publications.

Page 7. Line 20. Do you rather mean "evapotranspiration".

Table S7. If I understand correctly "L" should read "l".

[Figure]

---

## Referee Comment (RC2) · Anonymous Referee #2 · 27 Jun 2017

The manuscript "Pesticide fate at catchment scale: conceptual modelling of stream CSIA data" by Lutz et al. presents a combined data-analysis and modelling study, exploring the potential of transit time-based formulations of conceptual hydrological models to reproduce pesticide dynamics on different scales. The experiment is well-designed – in particular the comparison of alternative model set-ups is of critical importance (cf. "hypotheses testing") – and based on sound methods as far as hydrology is concerned (note that I am not an expert in chemistry and I cannot therefore not really evaluate the validity of these aspects in the manuscript). The manuscript may be of interest to many in the community as it is a clear demonstration that even relatively parsimonious model frameworks have considerable potential to reproduce and predict non-conservative hydro-geochemical dynamics at the catchment scale. I only

have a few minor remarks and I would thus be glad to eventually see this manuscript published.

(1) P.3,l.7: "confirm" may be too strong a term, perhaps replace by "support"

(2) P.5, section 2.3/2.4: the number of samples taken is not entirely clear. Maybe I misunderstood something, but in line 6 it is stated that a sample was taken at the catchment outlet every 20m3 between 03 /2012 and 08/2012. In line 18 it is stated that 34 samples were available. 34 samples over a period of 6 months if sampled at 20m3 intervals does not seem a lot, even if it is a very small catchment. Please check and clarify.

(3) P.6, section 2.5: it is not completely clear how or if pesticide uptake by plants was considered (essentially a loss term). Obviously it is desirable that there is no plant uptake of pesticides in reality. But is it so? Can this assumption be justified? Other authors seem to imply otherwise (e.g. Fantke et al., 2011, Chemosphere) and also Figure 2 in the manuscript seems to include a pesticide flux into vegetation. Yet, I could not find this reflected in any of the equations. Please clarify.

(4) P.6, section 2.5: please provide more information about the time-variant formulation of the SAS function. How was this done? Which type of distribution was chosen? Which parameter ranges were chosen and thus which shapes were possible?

(5) P.6, section 2.5, Figure 2: the energy input and/or potential evaporation is missing as incoming flux in figure 2

(6) P.6, section 2.5, l.23: Hrachowitz et al. (2015, Hydrological Processes) would fit better here.

(7) P.7, section 2.6 and 2.7: it is stated that pesticides are mostly applied during dry periods and that drying leads to particle adsorption to soil particles. The study site description suggests that the soils are mostly silty-clay. While in section 2.7 volatilization and deposition is mentioned, I can imagine that in addition wind induced migration of

soil particles will lead to some degree of pesticide redistribution (i.e. deposition minus erosion), in particular on arable land. This is obviously difficult to quantify, but may warrant some discussion.

(8) P.7, section 2.6, Table S6: I think it may be clearer to provide the equation for plant exudation in the following form to avoid confusion: phiex(t)=fex*phiet(t)

(9) P.7, l.20ff: I am not entirely convinced that this reasoning makes sense. What is the source zone? In most "conceptual" hydrological models it is the part of unsaturated zone that contributes to the non-linear response of hydrological systems. Roughly speaking, this is due to the fact that storage capacities below field capacity are generated by (1) soil evaporation and more importantly by (2) plants extracting water with their roots for transpiration. This essentially implies that the source zone encompasses the unsaturated root zone. As in deeper layers (i.e. "transport zone"), direct soil evaporation becomes of less importance and, by definition, no roots are present anymore (as it is not the root zone anymore) and thus the water content is always close to field capacity (except for the moments when a wetting front passes), the presence of a significant upward flux caused by evaporation or transpiration is rather unlikely. I believe that the conceptualization of ETtz and the associated phiet should be reconsidered. Although it is, of course, clearly possible (if not even likely) that there is an upward flux, I think it will be, given the fine grained soils, either be linked to capillary rise, or, what I find most plausible given my limited knowledge of the study site, is that these upward water and pesticide fluxes are linked to fluctuations in the groundwater table (i.e. the changing depth of the source and transport zones, respectively), reflecting a bit what was reported by Rouxel et al. (2011, Hydrological Processes).

(10) P.8, section 2.8: the calibration and model evaluation procedure would benefit from some more detail. Was the model *simultaneously* calibrated with respect to the three objective function, or only with respect to one of them, or individually one after the other? If simultaneously, how were the individual objective functions weighted? Which model performance was accepted as behavioural? What was used as likelihood

weight for the uncertainty estimation? In addition, please do not only provide the prior parameter distributions (Table S7) but also the posterior distributions. Also, given that the source zone storage capacity essentially reflects the storage capacity in the unsaturated root zone, a value between 0.1 and 10mm (Table S7) seems to be excessively low for this not very humid environment (i.e. aridity index ∼1.2). For such an environment this storage capacity is more likely to be in the range of about 50-250mm as recently suggested by Gao et al. (2014, Geophysical Research Letters).

(11) P.9, section 3.1, Figure 3: please add flow and/or precipitation to Figure 3 to allow the reader to make the link between water and pesticide dynamics.

(12) P.12, section 3.3: although nicely discussed and presented in Table 1, it may be interesting to see how/if the individual relative contributions change over time. I would be glad to see a figure showing that.

(13) P.13, section 3.4, l.12-15: please provide a bit more detail here. How was this assessment made? On basis of the model performance for the calibration period? Or post-calibration in a validation period? This is a crucial difference: if the assessment was done based on the calibration period, it is not at all surprising that a model with more calibration parameters (and thus more degrees of freedom) provides a better performance. It is almost (accounting for the uncertainties in the low number of Monte Carlo realizations used in the model) a mathematical necessity and thus provides only limited information about the model improvement. This can only be done in a meaningful way if compared for an independent test period (i.e. "validation period"). Please clarify.

(14) a more general remark: the similarity check indicated a relatively high overlap with previously published material (PhD-thesis?). You may want to reformulate the relevant parts of the manuscript to avoid complications.

---

## Editor Comment (EC1) · B. Schaefli (Editor) · 27 Jun 2017

Both reviewers agree that this manuscript is of high quality and that it makes a significant contribution to the literature. I would like to invite the authors to answer the detailed suggestions on how to improve the quality of the manuscript before revising their manuscript.

I agree with reviewer 1 that a redistribution of the material between the main text and the supplementary information will increase the readability of the paper and that the math notations should be improved. I also would like to recall that HESS does not like to have "double-variable" names such as "NS" (see author instructions).

[Figure]

202, 2017.

---

## Author Comment (AC1) · 27 Jun 2017

The authors decided to put most of the model technical details in the supplementary material, which is a viable option. However, the description in the main text does not stand alone and the reader is forced to go back and forth between the text and the Supplementary Tables. So I suggest to either put an even more concise version of the methods in the text (and develop a more detailed version in the SM), or put the equation and parameter description in the main text.

Reply: We thank the reviewer for the appreciation of our work and the useful and valuable comments. We acknowledge that the current layout requires the reader to go forth and back between the SM and the main text. Hence, we will follow the reviewer's

suggestion by moving the tables S5–7 from the SM to the main text.

Specific comments

1. I had a hard time following the equations in Table S5 because there is a mixture of continuous (differential equations) and finite difference equations. The author should decide one way to present the model and stick with it. I would suggest to use a continuous formulation. How this is then discretized into a finite difference equation for the numerical evaluation is quite trivial.

Reply: We agree with the reviewer that the mixture of differential and finite difference equations is confusing. We will revise all equations by using the continuous formulation as suggested by the reviewer. Moreover, we will remove "(t)" from all the equations (except for the ones for Csz, Ctz, CET) to improve the readability of the equations.

2. Table S5: Rmax seems like a maximum recharge rate, but please note that the description is missing from Table S7. Moreover, when Qsz > Rmax, where is the remaining flux going? In this equation, I was expecting to see Qsz - OF, to account for the fraction of Qsz not going to recharge the tz. Something is unclear in these equations, please clarify.

Reply: The parameter Rmax is indeed missing in Table S7 as it stems from a former implementation of recharge where the maximum recharge rate was set constant. In the current model formulation, the infiltration capacity of the transport zone is specified by a normal distribution (cf. page 6, lines 14-15 of the main text). Hence, in the revised version, we will change the equation for recharge to the transport zone to Rtz = Qsz – OF to account for overland flow, i.e., the outflow from the source zone that does not flow into the transport zone.

3. Table S6. C0(t) is computed assuming a well mixed reactor (i.e. total mass divided by storage). However, this seems to contradict the model formulation which assumes that every parcel of water has a certain pesticide concentration that depends on the

age, and the age distribution differs from the well mixed one. The rational for this choice must be explained.

Reply: $C_0(t)$ refers to the average concentration in the sorbed phase of the source zone, which is, indeed, set to the total mass divided by storage. However, the concentration in the source zone outflow does depend on the age distribution of the outflow, which is implemented in the equation of $C_{sz}(t)$ by using $p_{Q,sz}(T_{sz},t)$. In other words, the dissolved phase of the source zone does not behave as a well-mixed reactor and thus discharges pesticide molecules with various ages.

4. Page 9, lines 6-9. I read these lines a couple of time but I could not figure out exactly what was actually done. Which algorithm did you use for calibration. How large was the NS-efficiency? And the NS-efficiency range? Please expand and clarify on this. With 18 free parameters, the calibration is always going to be a critical point.

Reply: We agree with the reviewer that the calibration procedure should be clarified, which will be added to section 2.8 in the revised manuscript. Briefly, we calibrated the model against the combined objective function NScomb= $(1/6*NSQ+NSC+NS\delta13C)/(13/6)$ using the NSQ, NSC, and NS$\delta$13C coefficients as described in the SM. NScomb thus prioritises measured concentrations and $\delta$13C-values over measured discharge (see factor 1/6, which was determined in prior test calibration runs).

We applied the particle swarm optimization algorithm implemented in the open-source R package "HydroPSO" (Zambrano-Bigiarini and Rojas, 2013) and considered parameter sets behavioural if NScomb$\geq$0.7. This criterion was used to determine 10,000 behavioural parameter sets. The NS-efficiency of these behavioural parameter sets ranged between NScomb = 0.7 and NScomb = 0.92 (mean of 0.88), which will be mentioned in section 3.2 in the revised manuscript.

5. The authors should show the distribution of the "behavioral parameters". Were the parameters identifiable? With such a high parameter vs data ratio, I am expecting a

quite broad distribution. This should help explaining why the model could be calibrated reasonably well also without degradation.

Reply: The parameter identifiability is, indeed, a crucial aspect for conceptual hydrological models such as ours. We will show the distribution of behavioural parameters (see uploaded Fig. 1) together with a discussion on parameter identifiability in the SM and comment on it in section 3.2 in the revised text. Most model parameters show one clear maximum in the frequency distributions, apart from two flow-related and two pesticide model parameters, respectively. The two parameters with a limited identifiability in the flow model are those defining the SAS functions for ET ($\alpha$ET) and old water in discharge from the transport ($\beta$Q), respectively. The pesticide model shows a limited identifiability for the parameter determining pesticide transport in ET from the transport zone to the source zone (fex), as well as for the calibration factor of the applied pesticide mass (mIN). Hence, based on the measured data, it was not possible to distinguish the effects of ET from the effects of old water discharge on pesticide concentrations in the study catchment.

Overall, with 14 parameters showing distinct maxima in the histograms, we consider the amount of parameters reasonable in view of the variety of processes described in the model (e.g., time-varying storage selection, and different pesticide degradation and transport processes). Please note that the model did not calibrate well against measured pesticide concentrations without degradation (see Fig. S1 in the SM), which indicates that the concentration reduction at the catchment outlet cannot be ascribed to dilution only.

6. I would anticipate in the model description that some assumptions will be relaxed later, as shown in the result section. Otherwise the reader would continue reading wondering whether all the complexity is really necessary.

Reply: As discussed in section 3.4 in the main text ("Insights on pesticide fate and transport from the model"), the alternative model setups did not improve the represen-

tation of pesticide transport and degradation. Therefore, the original model setup as described in section 2.6 was kept and no model assumptions were relaxed. As this might not have become clear enough, we will specifically state in section 3.4 that the different alternative models tested were not adapted due to lower performance and larger uncertainties compared to the original (i.e., final) model.

Minor comments

1. Page 2. Line 22: I would move "provided that ... non-toxic" at the end of the sentence.

Reply: According to the reviewer's suggestion, we will move this part to the end of the sentence.

2. Page 7. Line 13. Is this type of modeling of desorption introduced here for the first time? If so, please expand a little the description. Otherwise refer to other publications.

Reply: This type of desorption kinetics has been introduced before in the modelling of nitrate, where a clear dilution effect during storms was found because of nitrate retention in the topsoil (van der Velde, 2010). This reference will be added to the revised version of the manuscript.

As mentioned in section 2.7, we assume that applied pesticides are largely retained in the sorbed phase rather than in the dissolved phase, as farmers will use pesticides preferably during dry periods to prevent losses via fast runoff. Hence, water in the applied spray formulation will quickly evaporate, leaving the pesticide sorbed to the soil and plants.

3. Page 7. Line 20. Do you rather mean "evapotranspiration".

Reply: This is indeed a typographical error, which will be corrected in the revised manuscript.

4. Table S7. If I understand correctly "L" should read "l".
Reply: The reviewer is right. The "coefficient describing pesticide sorption in the source zone" will be referred to with a lowercase l in the revised manuscript.

References van der Velde, Y.; de Rooij, G. H.; Rozemeijer, J. C.; van Geer, F. C.; Broers, H. P., Nitrate response of a lowland catchment: On the relation between stream concentration and travel time distribution dynamics. Water Resour. Res. 2010, 46, W11534, doi:10.1029/2010WR009105.

Zambrano-Bigiarini, M.; Rojas, R., A model-independent Particle Swarm Optimisation software for model calibration. Environmental Modelling & Software 2013, 43, 5-25.

[Figure]

Fig. 1. Histograms (frequency distributions) of the 18 calibrated model parameters from the 10,000 behavioural model simulations.

---

## Author Comment (AC2) · 17 Jul 2017

Reply to Reviewer 2

General comments

The manuscript "Pesticide fate at catchment scale: conceptual modelling of stream CSIA data" by Lutz et al. presents a combined data-analysis and modelling study, exploring the potential of transit time-based formulations of conceptual hydrological models to reproduce pesticide dynamics on different scales. The experiment is well designed – in particular the comparison of alternative model set-ups is of critical importance (cf. "hypotheses testing") – and based on sound methods as far as hydrology is concerned (note that I am not an expert in chemistry and I cannot therefore not

really evaluate the validity of these aspects in the manuscript). The manuscript may be of interest to many in the community as it is a clear demonstration that even relatively parsimonious model frameworks have considerable potential to reproduce and predict non-conservative hydro-geochemical dynamics at the catchment scale. Reply: We thank the reviewer for the appreciation of our work and the useful and valuable comments.

Specific comments

1. P.3, l.7: "confirm" may be too strong a term, perhaps replace by "support"

Reply: In the revised manuscript, we will replace "confirm the occurrence of pesticide degradation" by "provide evidence of pesticide degradation".

2. P.5, section 2.3/2.4: the number of samples taken is not entirely clear. Maybe I misunderstood something, but in line 6 it is stated that a sample was taken at the catchment outlet every 20m3 between 03 /2012 and 08/2012. In line 18 it is stated that 34 samples were available. 34 samples over a period of 6 months if sampled at 20m3 intervals does not seem a lot, even if it is a very small catchment. Please check and clarify.

Reply: We agree with the reviewer that some information is missing here. Runoff water was sampled every 20 m3 and consecutive samples were then combined to composite samples, leading to a total of 34 samples in six months. During baseflow conditions, samples were merged into weekly composite samples, whereas during runoff events, samples were merged according to the hydrograph components (i.e., baseflow, rising and falling limb) into several composite samples. This information will be added to the revised manuscript.

3. P.6, section 2.5: it is not completely clear how or if pesticide uptake by plants was considered (essentially a loss term). Obviously it is desirable that there is no plant uptake of pesticides in reality. But is it so? Can this assumption be justified? Other

authors seem to imply otherwise (e.g. Fantke et al., 2011, Chemosphere) and also Figure 2 in the manuscript seems to include a pesticide flux into vegetation. Yet, I could not find this reflected in any of the equations. Please clarify.

Reply: Pesticide uptake by plants was accounted for indirectly, as only a fraction of the pesticide evaporated from the source zone is redirected back into source zone storage (see page 7, lines 20–23). The remainder is thus taken up by plants without re-entering the source zone via plant exudation eventually. The total pesticide mass in ET from the transport zone is determined as ETsz*CET. The model parameter fex (see Table S6, equation for mass flux via plant exudation) gives the fraction of the total mass that will not remain in plants, i.e., the net pesticide transport from the transport zone to the source zone via ET.

4. P.6, section 2.5: please provide more information about the time-variant formulation of the SAS function. How was this done? Which type of distribution was chosen? Which parameter ranges were chosen and thus which shapes were possible?

Reply: The SAS function was approximated by a beta distribution defined by the mixing parameter mQ(t) (cf. van der Velde et al., 2015). The latter depends on the model parameters $\alpha Q$ and $\beta Q$. The equation for mQ(t) is shown in the attached figure, where Smin and Smax are the minimum and maximum transport zone storage, respectively. The parameter $\alpha Q$ ranges between 0.2 and 1.9, and $\beta Q$ ranges between 0 and 0.95 (see Table S7). Under dry conditions, mQ(t) approaches $\alpha Q$ and will primarily lead to old water discharge, whereas, under wet conditions, mQ(t) approaches $\alpha Q(1- \beta Q)$ and will primarily lead to young water discharge. In other words, the SAS function results in preference for young water if mQ(t)<1, preference for old water if mQ(t)>1, and a uniform distribution if mQ(t)=1 (i.e., "random sampling" of outflow from storage; see Fig. 1 in van der Velde et al., 2015). The equation for mQ(t) will be added to the SM to provide more information on the SAS approach chosen.

5. P.6, section 2.5, Figure 2: the energy input and/or potential evaporation is missing

as incoming flux in figure 2

Reply: We will add solar radiation ("IS") as incoming energy input flux to Fig.1.

6. P.6, section 2.5, l.23: Hrachowitz et al. (2015, Hydrological Processes) would fit better here.

Reply: We will replace the reference to Hrachowitz et al. (2016, Water) by Hrachowitz et al. (2015, Hydrological Processes).

7. P.7, section 2.6 and 2.7: it is stated that pesticides are mostly applied during dry periods and that drying leads to particle adsorption to soil particles. The study site description suggests that the soils are mostly silty-clay. While in section 2.7 volatilization and deposition is mentioned, I can imagine that in addition wind induced migration of soil particles will lead to some degree of pesticide redistribution (i.e. deposition minus erosion), in particular on arable land. This is obviously difficult to quantify, but may warrant some discussion.

Reply: Pesticide redistribution by wind-induced erosion might be a significant process, which is, indeed, difficult to quantify. However, the role of this process in the study catchment is assumed minor relative to erosion via overland flow, which is accounted for in the pesticide model. This aspect will be added to the revised version of the manuscript at the end of section 2.7. Moreover, we will mention wind-induced erosion as potential reason for the detection of acetochlor in the plot samples, in addition to drift and applications in previous years.

8. P.7, section 2.6, Table S6: I think it may be clearer to provide the equation for plant exudation in the following form to avoid confusion: phiex(t)=fex*phiet(t).

Reply: We will change the expression for $\Phi$ex(t) accordingly in order to avoid confusion.

9. P.7, l.20ff: I am not entirely convinced that this reasoning makes sense. What is the source zone? In most "conceptual" hydrological models it is the part of unsaturated zone that contributes to the non-linear response of hydrological systems. Roughly

speaking, this is due to the fact that storage capacities below field capacity are generated by (1) soil evaporation and more importantly by (2) plants extracting water with their roots for transpiration. This essentially implies that the source zone encompasses the unsaturated root zone. As in deeper layers (i.e. "transport zone"), direct soil evaporation becomes of less importance and, by definition, no roots are present anymore (as it is not the root zone anymore) and thus the water content is always close to field capacity (except for the moments when a wetting front passes), the presence of a significant upward flux caused by evaporation or transpiration is rather unlikely. I believe that the conceptualization of ETtz and the associated phiet should be reconsidered. Although it is, of course, clearly possible (if not even likely) that there is an upward flux, I think it will be, given the fine grained soils, either be linked to capillary rise, or, what I find most plausible given my limited knowledge of the study site, is that these upward water and pesticide fluxes are linked to fluctuations in the groundwater table (i.e. the changing depth of the source and transport zones, respectively), reflecting a bit what was reported by Rouxel et al. (2011, Hydrological Processes).

Reply: In our model, the source zone is a shallow layer at the ground surface, where the applied pesticide is initially sorbed and flushed out by infiltrating water (cf. Bertuzzo et al., 2013). Hence, the term "source zone" refers to the source of pesticide rather than the source of water. The transport zone comprises the entire subsurface below this shallow layer, i.e., the unsaturated zone including the root zone, and the aquifer. Hence, evapotranspiration from the transport zone needs to be simulated. Instead of further compartmentalising the subsurface, we opted for a single control volume and implemented time-varying storage selection to produce "non-random" sampling from storage (cf. the "direct SAS approach" in Benettin et al., 2017).

We will remove the plant symbols in Fig. 2, as they might erroneously suggest that the root zone does not extend to the transport zone.

10. P.8, section 2.8: the calibration and model evaluation procedure would benefit from some more detail. Was the model *simultaneously* calibrated with respect to the

three objective function, or only with respect to one of them, or individually one after the other? If simultaneously, how were the individual objective functions weighted? Which model performance was accepted as behavioural? What was used as likelihood weight for the uncertainty estimation? In addition, please do not only provide the prior parameter distributions (Table S7) but also the posterior distributions. Also, given that the source zone storage capacity essentially reflects the storage capacity in the unsaturated root zone, a value between 0.1 and 10mm (Table S7) seems to be excessively low for this not very humid environment (i.e. aridity index $\sim$ 1.2). For such an environment this storage capacity is more likely to be in the range of about 50-250mm as recently suggested by Gao et al. (2014, Geophysical Research Letters).

Reply: We agree with the reviewer that the calibration and model evaluation should be clarified, which we will do in section 2.8 in the revised manuscript. Briefly, we calibrated the model simultaneously against the NSQ, NSC, and NS$\delta$13C coefficients by using the combined objective function NScomb= (1/6*NSQ+NSC+NS$\delta$13C)/(13/6). The factor 1/6, which was determined via prior test calibration runs, ensures that all three terms contribute approximately evenly during the optimization process. The equation for NScomb will be added to the SM.

We applied the particle swarm optimization algorithm implemented in the open-source R package "HydroPSO" (Zambrano-Bigiarini and Rojas, 2013) and considered parameter sets behavioural if NScomb$\geq$0.7. This criterion was used to determine 10,000 behavioural parameter sets. The NS-efficiency of these behavioural parameter sets ranged between NScomb = 0.7 and NScomb = 0.92 (mean of 0.88), which will be mentioned in section 3.2 in the revised manuscript. The posterior parameter distributions will be shown and briefly discussed in an additional figure in the revised SM.

As the source zone represents the upmost soil layer at the ground surface where the pesticide is applied, we assume that a maximum storage capacity of 10 mm is sufficient. If the source zone represented the entire root zone, this value would, indeed, be too small. As explained above, we tried to minimize the compartmentalisation of catchment storage, which also avoids additional parameters to define the storage capacity of each catchment compartment.

11. P.9, section 3.1, Figure 3: please add flow and/or precipitation to Figure 3 to allow the reader to make the link between water and pesticide dynamics.

Reply: Precipitation and discharge time series will be added at the top of both columns of Figure 3 in the revised manuscript.

12. P.12, section 3.3: although nicely discussed and presented in Table 1, it may be interesting to see how/if the individual relative contributions change over time. I would be glad to see a figure showing that.

Reply: A figure showing the contribution of the mass-balance terms in 2012 will be added and referred to in the revised version of the manuscript.

13. P.13, section 3.4, l.12-15: please provide a bit more detail here. How was this assessment made? On basis of the model performance for the calibration period? Or post-calibration in a validation period? This is a crucial difference: if the assessment was done based on the calibration period, it is not at all surprising that a model with more calibration parameters (and thus more degrees of freedom) provides a better performance. It is almost (accounting for the uncertainties in the low number of Monte Carlo realizations used in the model) a mathematical necessity and thus provides only limited information about the model improvement. This can only be done in a meaningful way if compared for an independent test period (i.e. "validation period"). Please clarify.

Reply: Unfortunately, due to the limited amount of measured data, the comparison of the different model setups was not possible for a validation period. We fully agree with the reviewer that a more detailed model should always improve the model results during calibration. Therefore, indeed, the observation that the model improves by itself is not that valuable. However, because we implemented several small model adjustments,

we can compare the relative change in NSE between the alternative model setups. Furthermore, we compared the model results range outside the calibration period. We argue that if the more detailed model yields a smaller range in model results outside the calibration period compared to the range of the simpler model during the same period, the more detailed model is actually an improved model that is better able to grasp the flow and transport processes. In contrast, if the result range had been larger for the more detailed model, this would have indicated that the extra parameters mostly led to an increased model equifinality and thus did not really improve the model.

14. A more general remark: the similarity check indicated a relatively high overlap with previously published material (PhD-thesis?). You may want to reformulate the relevant parts of the manuscript to avoid complications.

Reply: The reviewer is right that parts of the manuscript are based on a chapter of the PhD-thesis by the main author. Despite the high overlap indicated by the similarity check, the manuscript has been considerably changed and improved with respect to the thesis chapter. We were in the understanding that self-plagiarism is not applicable in the case of material transferred between a PhD-thesis and respective journal papers of the same author. We checked with the editorial office of HESS, which confirmed that using parts of a PhD-thesis text without rephrasing is permitted. The thesis chapter has been published on the university's website as part of a PhD-thesis, but not in a scientific journal. Given the answer of the editorial office, we thus refrain from rephrasing the similar parts in the manuscript.

References

Benettin, P., C. Soulsby, C. Birkel, D. Tetzlaff, G. Botter, and A. Rinaldo (2017), Using SAS functions and high-resolution isotope data to unravel travel time distributions in headwater catchments, Water Resour. Res., 53, 1864–1878, doi:10.1002/2016WR020117.

Bertuzzo, E.; Thomet, M.; Botter, G.; Rinaldo, A., Catchment-scale herbicides transport: Theory and application. Advances in Water Resources 2013, 52, (0), 232-242, doi:10.1016/j.advwatres.2012.11.007.

van der Velde, Y.; Heidbüchel, I.; Lyon, S. W.; Nyberg, L.; Rodhe, A.; Bishop, K.; Troch, P. A., Consequences of mixing assumptions for time-variable travel time distributions. Hydrological Processes 2015, 29, (16), 1099-1085, doi:10.1002/hyp.10372.

$$m_Q(t) = \alpha_Q \left( 1 - \beta_Q \frac{S(t) - S_{min}}{S_{max} - S_{min}} \right)$$

**Fig. 1.**

---

## Author Comment (AC3) · 4 Aug 2017

We thank the editor for her appreciation of our work and the invitation to revise our manuscript.

We have already replied to all reviewer comments and will consider those in the revision. We will improve the mathematical notations as well as the distribution of the material between main text and supplementary material. We will also avoid multi-letter variables in the revised manuscript.

We will submit the revised manuscript in due time.

202, 2017.

---

## Editor Comment (EC2) · B. Schaefli (Editor) · 7 Aug 2017

The reviewer raised the question whether the material taken from a previously published PhD thesis should be reformulated (rather than copied literally).

To date, the HESS publication policy does not explicitly address this question. While self-plagiarism is not acceptable in general, the re-use of text pieces from the published PhD thesis is usually considered as the single acceptable exception.

As written in the author's response, the executive editor, E. Zehe, explicitly stated in his e-mail to the authors (17.7.2017) that the author "may literally use the text from [her] Phd thesis withouth rephrasing."

---

## Author Response (AR1)

**Author's response**

**1. Reply to Reviewer 1**

**General comments**

This manuscript combines field measurements (pesticide concentration and isotopic composition) with modeling to investigate the fate of pesticides in the hydrological cycle. The modeling part uses a state-of-the-art approach that features description of water ages using state-dependent SAS functions.

Overall, the MS makes a significant contribution to the literature on this topic by extending current models to the analysis of compound specific stable isotopes (CSIA). Moreover, it represents a good example of how data and models should be combined to gain the maximum knowledge of the underlying processes. There are however some issues that need to be addressed before publication.

The authors decided to put most of the model technical details in the supplementary material, which is a viable option. However, the description in the main text does not stand alone and the reader is forced to go back and forth between the text and the Supplementary Tables. So I suggest to either put an even more concise version of the methods in the text (and develop a more detailed version in the SM), or put the equation and parameter description in the main text.

**Reply:** We thank the reviewer for the appreciation of our work and the useful and valuable comments. We acknowledge that the current layout requires the reader to go forth and back between the SM and the main text. Hence, we followed the reviewer's suggestion by moving the tables S5–7 from the SM to the main text (Tables 1–3 in the revised manuscript).

**Specific comments**

1. I had a hard time following the equations in Table S5 because there is a mixture of continuous (differential equations) and finite difference equations. The author should

   decide one way to present the model and stick with it. I would suggest to use a continuous formulation. How this is then discretized into a finite difference equation for the numerical evaluation is quite trivial.

**Reply:** We agree with the reviewer that the mixture of differential and finite difference equations is confusing. We revised all equations by using the continuous formulation as suggested by the reviewer (see Tables 1 and 2 in the revised manuscript). Moreover, we removed "(t)" from all equations (except for the ones for $C_{sz}$, $C_{tz}$ and $C_{ET}$) to improve the readability of the equations.

2. Table S5: $R_{max}$ seems like a maximum recharge rate, but please note that the description

   is missing from Table S7. Moreover, when $Q_{sz} > R_{max}$, where is the remaining flux going? In this equation, I was expecting to see $Q_{sz}$ - OF, to account for the fraction of $Q_{sz}$ not going to recharge the tz. Something is unclear in these equations, please clarify.

**Reply:** The parameter $R_{max}$ is indeed missing in Table S7 as it stems from a former implementation of recharge where the maximum recharge rate was set constant. In the current model formulation, the infiltration capacity of the transport zone is specified by a normal distribution (cf. page 6, lines 14-15 of the original manuscript). Hence, in the revised version, we changed the equation for recharge to the transport zone to $R_{tz} = Q_{sz} - OF$ (see Table 1) to account for overland flow, i.e., the outflow from the source zone that does not flow into the transport zone.

3. Table S6. $C_0(t)$ is computed assuming a well mixed reactor (i.e. total mass divided by storage). However, this seems to contradict the model formulation which assumes that every parcel of water has a certain pesticide concentration that depends on the age, and the age distribution differs from the well mixed one. The rational for this choice must be explained.

**Reply:** $C_0(t)$ refers to the average concentration in the sorbed phase of the source zone, which is, indeed, set to the total mass divided by storage. However, the concentration in the source zone outflow does depend on the age distribution of the outflow, which is implemented in the equation of $C_{sz}(t)$ by using $p_{Q,sz}(T_{sz},t)$. In other words, the dissolved phase of the source zone does not behave as a well-mixed reactor and thus discharges pesticide molecules with various ages.

4. Page 9, lines 6-9. I read these lines a couple of time but I could not figure out exactly what was actually done. Which algorithm did you use for calibration. How large was the NS-efficiency? And the NS-efficiency range? Please expand and clarify on this. With 18 free parameters, the calibration is always going to be a critical point.

**Reply**: We agree with the reviewer that the calibration procedure should be clarified, which was done in section 2.8 in the revised manuscript (page 9, lines 10–16). Briefly, we calibrated the model against the combined objective function $N_{comb}= (1/6*NS_Q+NS_C+NS_{\delta13C})/(13/6)$ using the $NS_Q$, $NS_C$, and $NS_{\delta13C}$ coefficients as described in the SM (now $N_Q$, $N_C$, and $N_{\delta13C}$ to avoid multi-letter variables). The factor 1/6 for $NS_Q$ was determined through prior test calibration runs to ensure that all three terms contribute approximately evenly during the optimization process. The equation for $N_{comb}$ was added to section S5 in the SM.

We applied the particle swarm optimization algorithm implemented in the open-source R package "HydroPSO" (Zambrano-Bigiarini and Rojas, 2013) and considered parameter sets behavioural if $N_{comb}\geq0.7$. This criterion was used to determine 10,000 behavioural parameter sets. The NS-efficiency of these behavioural parameter sets ranged between $N_{comb} = 0.7$ and $N_{comb} = 0.92$ (mean of 0.88), which was added to section 3.2 in the revised manuscript (page 12, line 25).

5.  The authors should show the distribution of the "behavioral parameters". Were the parameters identifiable? With such a high parameter vs data ratio, I am expecting a quite broad distribution. This should help explaining why the model could be calibrated reasonably well also without degradation.

**Reply:** The parameter identifiability is, indeed, a crucial aspect for conceptual hydrological models such as ours. We included in the revised version of the SM a figure showing the distribution of behavioural parameters (Fig. S1) together with a discussion on parameter identifiability (see section S5), and commented on this in section 3.2 (page 12, lines 26–33) in the revised main text.

Most model parameters show one clear maximum in the frequency distributions, apart from two flow-related and two pesticide model parameters, respectively. The two parameters with a limited identifiability in the flow model are those defining the SAS functions for ET ($\alpha_{ET}$) and old water in discharge from the transport zone ($\beta_Q$), respectively. The pesticide model shows a limited identifiability for the parameter determining pesticide transport in ET from the transport zone to the source zone ($f_{ex}$), as well as for the calibration factor of the applied pesticide mass ($m_{IN}$). Hence, based on the measured data, it was not possible to distinguish the effects of ET from the effects of old water discharge on pesticide concentrations in the study catchment.

Overall, with 14 parameters showing distinct maxima in the histograms, we consider the amount of parameters reasonable in view of the variety of processes described in the model (e.g., time-varying storage selection, and different pesticide degradation and transport processes). Please

note that the model did not calibrate well against measured pesticide concentrations without degradation (see Fig. S1 in the original SM; now Fig. S2), which indicates that the concentration reduction at the catchment outlet cannot be ascribed to dilution only.

**6.** I would anticipate in the model description that some assumptions will be relaxed later, as shown in the result section. Otherwise the reader would continue reading wondering whether all the complexity is really necessary.

**Reply:** As discussed in section 3.4 in the main text ("Insights on pesticide fate and transport from the model"), the alternative model setups did not improve the representation of pesticide transport and degradation. Therefore, the original model setup as described in sections 2.6 and 2.7 was kept and no model assumptions were relaxed. As this might not have become clear enough, we specifically stated in section 3.4 of the revised manuscript (page 14, lines 22–24) that the different alternative models tested were not adapted due to lower performance and larger uncertainties compared to the original (i.e., final) model.

**Minor comments**

**1.** Page 2. Line 22: I would move "provided that ... non-toxic" at the end of the sentence.

**Reply:** According to the reviewer's suggestion, we moved this part to the end of the sentence (page 2, line 20).

**2.** Page 7. Line 13. Is this type of modeling of desorption introduced here for the first time? If so, please expand a little the description. Otherwise refer to other publications.

**Reply:** This type of desorption kinetics has been introduced before in the modelling of nitrate, where a clear dilution effect during storms was found because of nitrate retention in the topsoil (van der Velde, 2010). This reference was added to the revised version of the manuscript (page 7, line 14).
As mentioned in section 2.7 of the original manuscript, we assume that applied pesticides are largely retained in the sorbed phase rather than in the dissolved phase, as farmers will use pesticides preferably during dry periods to prevent losses via fast runoff. Hence, water in the applied spray formulation will quickly evaporate, leaving the pesticide sorbed to the soil and plants.

**3.** Page 7. Line 20. Do you rather mean "evapotranspiration".

**Reply:** This is indeed a typographical error, which was corrected in the revised manuscript (page 7, line 23).

**4.** Table S7. If I understand correctly "L" should read "l".

**Reply:** The reviewer is right. The "coefficient describing pesticide sorption in the source zone" was given a lowercase "l" symbol in the revised manuscript (Table 3 in the revised manuscript).

**2. Reply to Reviewer 2**

**General comments**

The manuscript "Pesticide fate at catchment scale: conceptual modelling of stream CSIA data" by Lutz et al. presents a combined data-analysis and modelling study, exploring the potential of transit time-based formulations of conceptual hydrological models to reproduce pesticide dynamics on different scales. The experiment is well designed – in particular the comparison of alternative model set-ups is of critical importance (cf. "hypotheses testing") – and based on sound methods as far as hydrology is concerned (note that I am not an expert in chemistry and I cannot therefore not really evaluate the validity of these aspects in the manuscript). The manuscript may be of interest to many in the community as it is a clear demonstration that even relatively parsimonious model frameworks have considerable potential to reproduce and predict non-conservative hydro-geochemical dynamics at the catchment scale.

**Reply:** We thank the reviewer for the appreciation of our work and the useful and valuable comments.

**Specific comments**

**1.** P.3, l.7: "confirm" may be too strong a term, perhaps replace by "support"

**Reply**: In the revised manuscript, we replaced "confirm the occurrence of pesticide degradation" by "provide evidence of pesticide degradation" (page 3, line 7).

2. P.5, section 2.3/2.4: the number of samples taken is not entirely clear. Maybe I misunderstood something, but in line 6 it is stated that a sample was taken at the catchment outlet every 20m3 between 03 /2012 and 08/2012. In line 18 it is stated that 34 samples were available. 34 samples over a period of 6 months if sampled at 20m3 intervals does not seem a lot, even if it is a very small catchment. Please check and clarify.

**Reply**: We agree with the reviewer that some information is missing here. Runoff water was sampled every 20 $m^3$ and consecutive samples were then combined to composite samples, leading to a total of 34 samples in six months. During baseflow conditions, samples were merged into weekly composite samples, whereas during runoff events, samples were merged into several composite samples according to the hydrograph components (i.e., baseflow, rising and falling limb). This information was added to section 2.3 in the revised manuscript (page 5, lines 7–9).

3. P.6, section 2.5: it is not completely clear how or if pesticide uptake by plants was considered (essentially a loss term). Obviously it is desirable that there is no plant uptake of pesticides in reality. But is it so? Can this assumption be justified? Other authors seem to imply otherwise (e.g. Fantke et al., 2011, Chemosphere) and also Figure 2 in the manuscript seems to include a pesticide flux into vegetation. Yet, I could not find this reflected in any of the equations. Please clarify.

**Reply**: Pesticide uptake by plants was accounted for indirectly, as only a fraction of the pesticide evaporated from the source zone is redirected back into source zone storage (see page 7, lines 21–23 in the original manuscript). The remainder is thus taken up by plants without re-entering the source zone via plant exudation eventually. The total pesticide mass in ET from the transport zone is determined as $E_{sz}*C_{ET}$. The model parameter $f_{ex}$ gives the fraction of the total mass that will not remain in plants, i.e., the net pesticide transport from the transport zone to the source zone via ET (see equation for mass flux via plant exudation in Table S6 of the original SM or Table 2 in the revised manuscript, respectively).

**4.** P.6, section 2.5: please provide more information about the time-variant formulation of the SAS function. How was this done? Which type of distribution was chosen? Which parameter ranges were chosen and thus which shapes were possible?

**Reply**: The SAS function was approximated by a beta distribution defined by the mixing parameter $m_Q(t)$ (cf. van der Velde et al., 2015). The latter depends on the model parameters $\alpha_Q$ and $\beta_Q$:

$$m_Q(t) = \alpha_Q \left(1 - \beta_Q \frac{S(t) - S_{min}}{S_{max} - S_{min}}\right)$$

where $S_{min}$ and $S_{max}$ are the minimum and maximum transport zone storage, respectively. The parameter $\alpha_Q$ ranges between 0.2 and 1.9, and $\beta_Q$ ranges between 0 and 0.95 (see Table S7 in the original SM or Table 3 in the revised manuscript, respectively). Under dry conditions, $m_Q(t)$ approaches $\alpha_Q$ and will primarily lead to old water discharge, whereas, under wet conditions, $m_Q(t)$ approaches $\alpha_Q(1- \beta_Q)$ and will primarily lead to young water discharge. In other words, the SAS function will have a preference for young water for $m_Q(t)<1$, a preference for old water for $m_Q(t)>1$ and represent a uniform distribution for $m_Q(t)=1$ (i.e., "random sampling" of outflow from storage; see Fig. 1 in van der Velde et al., 2015). This information was added to the revised version of the SM (new section S2) to provide more information on the SAS approach chosen.

**5.** P.6, section 2.5, Figure 2: the energy input and/or potential evaporation is missing as incoming flux in figure 2

**Reply**: We added solar radiation ("$R_S$") as incoming energy input flux to Fig.2 in the revised manuscript.

**6.** P.6, section 2.5, l.23: Hrachowitz et al. (2015, Hydrological Processes) would fit better here.

**Reply**: We replaced the reference to Hrachowitz et al. (2016, Water) by Hrachowitz et al. (2015, Hydrological Processes); see page 6, line 23.

**7.** P.7, section 2.6 and 2.7: it is stated that pesticides are mostly applied during dry periods and that drying leads to particle adsorption to soil particles. The study site description suggests that the soils are mostly silty-clay. While in section 2.7 volatilization and deposition is mentioned, I can imagine that in addition wind induced migration of soil particles will lead to some degree

of pesticide redistribution (i.e. deposition minus erosion), in particular on arable land. This is obviously difficult to quantify, but may warrant some discussion.

**Reply**: Pesticide redistribution by wind-induced erosion might be a significant process, which is, indeed, difficult to quantify. However, the role of this process in the study catchment is assumed minor relative to erosion via overland flow, which is accounted for in the pesticide model. This aspect was added to the revised version of the manuscript at the end of section 2.7 (page 8, lines 23–25). Moreover, we added wind-induced erosion as potential reason for the detection of acetochlor in the plot samples (page 10, lines 30–31) in the revised manuscript), in addition to drift and applications in previous years.

8. P.7, section 2.6, Table S6: I think it may be clearer to provide the equation for plant exudation in the following form to avoid confusion: phiex(t)=fex*phiet(t).

**Reply**: We changed the expression for $\Phi_{ex}(t)$ accordingly in order to avoid confusion (Table 2 in the revised manuscript).

9. P.7, l.20ff: I am not entirely convinced that this reasoning makes sense. What is the source zone? In most "conceptual" hydrological models it is the part of unsaturated zone that contributes to the non-linear response of hydrological systems. Roughly speaking, this is due to the fact that storage capacities below field capacity are generated by (1) soil evaporation and more importantly by (2) plants extracting water with their roots for transpiration. This essentially implies that the source zone encompasses the unsaturated root zone. As in deeper layers (i.e. "transport zone"), direct soil evaporation becomes of less importance and, by definition, no roots are present anymore (as it is not the root zone anymore) and thus the water content is always close to field capacity (except for the moments when a wetting front passes), the presence of a significant upward flux caused by evaporation or transpiration is rather unlikely. I believe that the conceptualization of ETtz and the associated phiet should be reconsidered. Although it is, of course, clearly possible (if not even likely) that there is an upward flux, I think it will be, given the fine grained soils, either be linked to capillary rise, or, what I find most plausible given my limited knowledge of the study site, is that these upward water and pesticide fluxes are linked to fluctuations in the groundwater table (i.e. the changing depth of the source and transport zones, respectively), reflecting a bit what was reported by Rouxel et al. (2011, Hydrological Processes).

**Reply**: In our model, the source zone is a shallow layer at the ground surface, where the applied pesticide is initially sorbed and flushed out by infiltrating water (cf. Bertuzzo et al., 2013). Hence, "source zone" refers to the source of pesticide rather than the source of water. The transport zone comprises the entire subsurface below this shallow layer, i.e., the unsaturated zone including the root zone, and the aquifer. Hence, evapotranspiration from the transport zone needs to be simulated. Instead of further compartmentalising the subsurface, we opted for a single control volume and implemented time-varying storage selection to produce "non-random" sampling from storage (cf. the "direct SAS approach" in Benettin et al., 2017).

We removed the plant symbols in Fig. 2 in the revised manuscript, as they might erroneously suggest that the root zone does not extend to the transport zone.

**10.** P.8, section 2.8: the calibration and model evaluation procedure would benefit from some more detail. Was the model \*simultaneously\* calibrated with respect to the three objective function, or only with respect to one of them, or individually one after the other? If simultaneously, how were the individual objective functions weighted? Which model performance was accepted as behavioural? What was used as likelihood weight for the uncertainty estimation? In addition, please do not only provide the prior parameter distributions (Table S7) but also the posterior distributions. Also, given that the source zone storage capacity essentially reflects the storage capacity in the unsaturated root zone, a value between 0.1 and 10mm (Table S7) seems to be excessively low for this not very humid environment (i.e. aridity index ~ 1.2). For such an environment this storage capacity is more likely to be in the range of about 50-250mm as recently suggested by Gao et al. (2014, Geophysical Research Letters).

**Reply**: We agree with the reviewer that the calibration and model evaluation should be clarified, which we did in section 2.8 in the revised manuscript (page 9, lines 10–16). Briefly, we calibrated the model simultaneously against the $NS_Q$, $NS_C$, and $NS_{\delta 13C}$ coefficients (now $N_Q$, $N_C$, and $N_{\delta 13C}$ to avoid multi-letter variables) by using the combined objective function $N_{comb}=$ $(1/6*NS_Q+NS_C+NS_{\delta 13C})/(13/6)$. The factor 1/6 was determined through prior test calibration runs to ensure that all three terms contribute approximately evenly during the optimization process. The equation for $N_{comb}$ was added to section S5 in the SM.

We applied the particle swarm optimization algorithm implemented in the open-source R package "HydroPSO" (Zambrano-Bigiarini and Rojas, 2013) and considered parameter sets behavioural if $N_{comb} \geq 0.7$. This criterion was used to determine 10,000 behavioural parameter sets. The NS-efficiency of these behavioural parameter sets ranged between $N_{comb} = 0.7$ and $N_{comb} = 0.92$ (mean of 0.88), which was mentioned in section 3.2 in the revised manuscript (page 12, line 25). The posterior parameter distributions are now shown and briefly discussed in the revised manuscript and SM (page 12, lines 26–33 in the manuscript; Fig. S1 and section S5 in the SM).

As the source zone represents the upmost soil layer at the ground surface where the pesticide is applied, we assume that a maximum storage capacity of 10 mm is sufficient. If the source zone represented the entire root zone, this value would, indeed, be too small. As explained above, we tried to minimize the compartmentalisation of catchment storage, which also avoids additional parameters to define the storage capacity of each catchment compartment.

**11.** P.9, section 3.1, Figure 3: please add flow and/or precipitation to Figure 3 to allow the reader to make the link between water and pesticide dynamics.

**Reply**: Precipitation and discharge time series were added to Figure 3 in the revised manuscript.

**12.** P.12, section 3.3: although nicely discussed and presented in Table 1, it may be interesting to see how/if the individual relative contributions change over time. I would be glad to see a figure showing that.

**Reply**: A figure showing the contribution of the mass-balance terms in 2012 was added and referred to in the revised version of the manuscript (new Fig. 5).

**13.** P.13, section 3.4, l.12-15: please provide a bit more detail here. How was this assessment made? On basis of the model performance for the calibration period? Or post-calibration in a validation period? This is a crucial difference: if the assessment was done based on the calibration period, it is not at all surprising that a model with more calibration parameters (and thus more degrees of freedom) provides a better performance. It is almost (accounting for the uncertainties in the low number of Monte Carlo realizations used in the model) a mathematical necessity and thus provides only limited information about the model improvement. This can only be done in a meaningful way if compared for an independent test period (i.e. "validation period"). Please clarify.

**Reply**: Unfortunately, due to the limited amount of measured data, the comparison of the two models was not possible for a validation period. We fully agree with the reviewer that a more detailed model should always improve the model results during calibration. Therefore, indeed, the observation that the model improves by itself is not that valuable. However, because we implemented several small model adjustments, we can compare the relative change in NSE between the alternative model setups. This was clarified in the revised manuscript by comparing the NSEs of the two simplified models (page 14, lines 7–10), instead of comparing the NSE of the original model to the NSE of each simplified model as done in the original manuscript. Furthermore, we compared the model results range for a year outside the calibration period. We argue that if the more detailed model yields a smaller range in model results outside the calibration period compared to the range of the simpler model during the same period, the more detailed model is actually an improved model that is better able to grasp the flow and transport processes. In contrast, if the result range had been larger for the more detailed model, this would have indicated that the extra parameters mostly led to an increased model equifinality and thus did not really improve the model.

**14.** A more general remark: the similarity check indicated a relatively high overlap with previously published material (PhD-thesis?). You may want to reformulate the relevant parts of the manuscript to avoid complications.

**Reply**: The reviewer is right that parts of the manuscript are based on a chapter of the PhD thesis by the main author. Despite the high overlap indicated by the similarity check, the manuscript has been considerably changed and improved with respect to the thesis chapter. We were in the understanding that self-plagiarism is not applicable in the case of material transferred between a PhD-thesis and respective journal papers of the same author. The executive editor E. Zehe confirmed that using parts of a PhD-thesis text without rephrasing is permitted (see also editor comment EC2). The thesis chapter has been published on the university's website as part of a PhD-thesis, but not in a scientific journal. Given the answer of the executive editor, we thus refrain from rephrasing the similar parts in the manuscript.

**3. References for reviewer replies**

Benettin, P., C. Soulsby, C. Birkel, D. Tetzlaff, G. Botter, and A. Rinaldo (2017), Using SAS functions and high-resolution isotope data to unravel travel time distributions in headwater catchments, Water Resour. Res., 53, 1864–1878, doi:10.1002/2016WR020117.

5  Bertuzzo, E.; Thomet, M.; Botter, G.; Rinaldo, A., Catchment-scale herbicides transport: Theory and application. Advances in Water Resources 2013, 52, (0), 232-242, doi:10.1016/j.advwatres.2012.11.007.

van der Velde, Y.; Heidbüchel, I.; Lyon, S. W.; Nyberg, L.; Rodhe, A.; Bishop, K.; Troch, P. A., Consequences of mixing assumptions for time-variable travel time distributions. Hydrological 10  Processes 2015, 29, (16), 1099-1085, doi:10.1002/hyp.10372.

van der Velde, Y.; de Rooij, G. H.; Rozemeijer, J. C.; van Geer, F. C.; Broers, H. P., Nitrate response of a lowland catchment: On the relation between stream concentration and travel time distribution dynamics. Water Resour. Res. 2010, 46, W11534, doi:10.1029/2010WR009105.

Zambrano-Bigiarini, M.; Rojas, R., A model-independent Particle Swarm Optimisation software for 15  model calibration. Environmental Modelling & Software 2013, 43, 5-25.

**4. Relevant changes in the revised manuscript**

1. We moved the tables S5–7 from the SM to the main text (Tables 1–3 in the revised manuscript).
2. We revised all model equations by using their continuous formulation (Table 1 in the revised manuscript).
3. We clarified the calibration procedure in section 2.8 of the revised manuscript (page 9, lines 10–16) and added the equation for the combined objective function $N_{comb}$ to the SM (see section S5).
4. To discuss the issue of parameter identifiability, we included in the revised version of the SM a figure showing the distribution of behavioural parameters (Fig. S1) together with a discussion on parameter identifiability (see section S5), and commented on this in section 3.2 in the revised manuscript (page 12, lines 26–33).
5. We corrected the references to studies using a constant degradation half-life for pesticides over the entire subsurface depth (page 14, line 19).
6. We specified the sampling frequency for water samples at the catchment outlet (section 2.3 in the revised manuscript; page 5, lines 7–9).
7. We provided more information on the SAS approach chosen in the revised version of the SM (new section S2).
8. A figure showing the contribution of the mass-balance terms to modelled pesticide degradation and transport in 2012 was added to the revised version of the manuscript (new Fig. 5).
9. We changed all multi-letter variables to single-letter variables as follows:
   a. $ET_{sz}$ to $E_{sz}$; $ET_{tz}$ to $E_{tz}$; $ET_{pot}$ to $E_{pot}$ (in sections 2.5 and 2.8, Tables 1 and 2 and Fig. 2 in the revised manuscript)
   b. $ED_{Rayleigh}$ to $D_{Rayleigh}$; $ED_{Sample}$ to $D_{Sample}$ (eqs. S3 and S4 in the revised SM; Fig. 7 in the revised manuscript)
   c. $NS_Q$ to $N_Q$; $NS_C$ to $N_C$; $NS_{\delta13C}$ to $N_{\delta13C}$ (eqs. S5–S7 in the revised SM; Fig. 4 in the revised manuscript)

[revised manuscript text omitted]